# Comprehensive substrate specificity profiling of the human Nek kinome reveals unexpected signaling outputs

Bert van de Kooij[1,2,3,4], Pau Creixell[1,2,3,4], Anne van Vlimmeren[1,2,3,4], Brian A Joughin[1,2,3,4], Chad J Miller[5†], Nasir Haider[6], Craig D Simpson[7], Rune Linding[7], Vuk Stambolic[6,9], Benjamin E Turk[5], Michael B Yaffe[1,2,3,4,8]*

[1]Department of Biology, Massachusetts Institute of Technology, Cambridge, United States; [2]Department of Biological Engineering, Massachusetts Institute of Technology, Cambridge, United States; [3]Koch Institute for Integrative Cancer Research, Massachusetts Institute of Technology, Cambridge, United States; [4]MIT Center for Precision Cancer Medicine, Massachusetts Institute of Technology, Cambridge, United States; [5]Department of Pharmacology, Yale School of Medicine, New Haven, United States; [6]Department of Medical Biophysics, University of Toronto, Toronto, Canada; [7]Biotech Research and Innovation Center, Faculty of Health and Medical Sciences, University of Copenhagen, Copenhagen, Denmark; [8]Department of Surgery, Beth Israel Deaconess Medical Center, Divisions of Acute Care Surgery, Trauma, and Critical Care and Surgical Oncology, Harvard Medical School, Boston, United States; [9]Princess Margaret Cancer Center, University Health Network, Toronto, Canada

*For correspondence:
myaffe@mit.edu

Present address: †Department of Biochemistry, University of Washington, Seattle, United States

Competing interests: The authors declare that no competing interests exist.

**Abstract** Human NimA-related kinases (Neks) have multiple mitotic and non-mitotic functions, but few substrates are known. We systematically determined the phosphorylation-site motifs for the entire Nek kinase family, except for Nek11. While all Nek kinases strongly select for hydrophobic residues in the −3 position, the family separates into four distinct groups based on specificity for a serine versus threonine phospho-acceptor, and preference for basic or acidic residues in other positions. Unlike Nek1-Nek9, Nek10 is a dual-specificity kinase that efficiently phosphorylates itself and peptide substrates on serine and tyrosine, and its activity is enhanced by tyrosine auto-phosphorylation. Nek10 dual-specificity depends on residues in the HRD+2 and APE-4 positions that are uncommon in either serine/threonine or tyrosine kinases. Finally, we show that the phosphorylation-site motifs for the mitotic kinases Nek6, Nek7 and Nek9 are essentially identical to that of their upstream activator Plk1, suggesting that Nek6/7/9 function as phospho-motif amplifiers of Plk1 signaling.
DOI: https://doi.org/10.7554/eLife.44635.001

## Introduction

Protein phosphorylation by kinases plays an essential role in nearly all signaling events and physiological processes within the cell. Protein serine/threonine kinases, together with ubiquitin ligases, play a particularly important role in mitosis, during which thousands of sites are phosphorylated (*Dephoure et al., 2008*). These mitotic phosphorylation events are predominantly mediated by the kinases Cdk1, Plk1, Aurora A and Aurora B, together with several members of the NimA-related kinase (Nek) family (*Fry et al., 2017*; *Joukov and De Nicolo, 2018*). For Cdk1, Plk1 and the Aurora kinases, multiple substrates and detailed downstream signaling functions have been identified

(*Joukov and De Nicolo, 2018*), although much remains to be learned. In contrast, for the Nek family kinases only a few substrates have been identified, and many of their functions within the complex network of mitotic signaling are not well understood.

The Nek kinase family consists of 11 serine/threonine kinases that together form an independent evolutionary branch of the human kinome (*Figure 1A*). The first human mitotic Nek kinase to be identified was Nek2, which is required for centrosomal disjunction following their duplication during S-phase, in order to assemble a bipolar mitotic spindle (*Figure 1B*; *Fry et al., 2017*) A similar function was assigned to Nek5, which, in addition, also regulates centrosome composition in interphase (*Prosser et al., 2015*). Furthermore, Nek6, Nek7 and Nek9 have been shown to cooperate as a signaling module during various stages of mitosis (*Belham et al., 2003*). In this process, Nek9 is first phosphorylated in early mitosis by Cdk1, creating a binding site for the Plk1 polo-box domain. Plk1-mediated phosphorylation and activation of Nek9 then leads to Nek9-dependent phosphorylation and interaction-dependent activation of Nek6 and Nek7, allowing them to perform their downstream functions (*Roig et al., 2002*; *Belham et al., 2003*; *Yin et al., 2003*; *Richards et al., 2009*; *Bertran et al., 2011*). As a signaling module, Nek6/7/9 have been reported to play a role in centrosome separation and maturation, nuclear envelope breakdown, metaphase and anaphase progression, mitotic spindle formation and cytokinesis (*Figure 1B*; *Roig et al., 2002*; *Yin et al., 2003*; *Yissachar et al., 2006*; *O'Regan and Fry, 2009*; *Salem et al., 2010*; *Bertran et al., 2011*; *Kim et al., 2011*; *Laurell et al., 2011*; *Sdelci et al., 2012*; *Cullati et al., 2017*). Interestingly, although Nek6 and Nek7 share more than 80% sequence homology in their kinase domain, their mitotic functions are not completely redundant (*Cullati et al., 2017*). This is most likely due to their short but divergent N-terminal domains that drive differential interactions and might affect substrate selection (*Vaz Meirelles et al., 2010*; *de Souza et al., 2014*).

Notably, the Nek kinase family is not exclusively involved in mitosis, but functionally diverse. Nek kinases have also been reported to play a role in meiosis, ciliary biology, and the response of cells to replication stress and DNA-damage (*Figure 1B*; *Quarmby and Mahjoub, 2005*; *Melixetian et al., 2009*; *Moniz and Stambolic, 2011*; *Choi et al., 2013*; *Spies et al., 2016*; *Brieño-Enríquez et al., 2017*). The functions of Nek3 and Nek4 are not clear but the current literature suggests that Nek3 may play a role in cell migration (*Harrington and Clevenger, 2016*), and Nek4 might be involved in regulating microtubule stability (*Figure 1B*; *Doles and Hemann, 2010*).

We have previously determined the optimal substrate phosphorylation motifs, that is the substrate amino acid sequence preferentially phosphorylated by a given kinase, for the mitotic kinases Cdk1, Plk1, Aurora A, Aurora B and Nek2 (*Alexander et al., 2011*). This motif information strongly facilitates identification of kinase substrates and can help to understand the organization of complex phosphorylation networks like those in mitotic cells (*Linding et al., 2007*; *Alexander et al., 2011*; *Kang et al., 2013*). Therefore, to further catalog the optimal substrate motif-atlas for mitotic kinases and to characterize substrate-specificity of a complete and functionally diverse branch of the human kinome, we sought to determine the optimal phosphorylation-site motif for the Nek-kinases using Oriented Peptide Library Screening (OPLS) (*Hutti et al., 2004*), followed by direct experimental validation together with molecular modeling. This revealed a surprisingly large amount of diversity in substrate-specificity among the Nek-kinases, and identified Nek10 as a dual-specificity kinase, that can phosphorylate substrates on tyrosine in addition to serine. Furthermore, these studies demonstrated that the mitotic kinases Nek6, Nek7 and Nek9, have a phosphorylation-site motif that is almost identical to the motif of Plk1, suggesting that the Nek6/7/9 module functions as a Plk1-motif amplifier.

## Results

### Determination of the phosphorylation-site motif for all members of the Nek kinase family

To obtain the consensus phosphorylation-site motif for each individual Nek kinase, we performed Oriented Peptide Library Screening (OPLS) as described previously (*Hutti et al., 2004*; *Alexander et al., 2011*). In these experiments, each kinase is tested in 198 individual *in vitro* kinase reactions, each with a different peptide library as substrate (*Figure 1—figure supplement 1A*). Each peptide library is composed of a pool of 10-mer peptides containing an equimolar mix of serine and

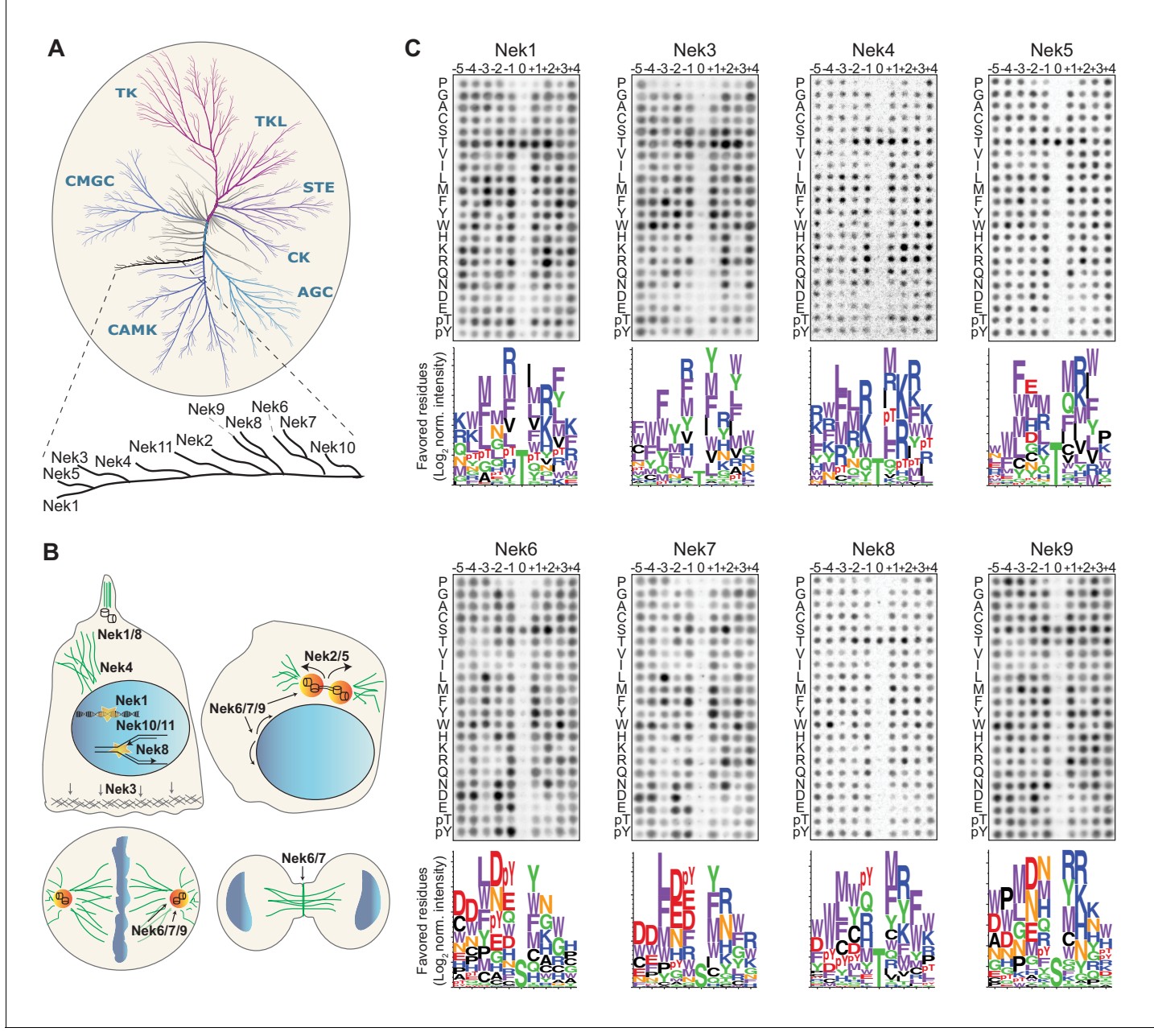

**Figure 1.** Phosphorylation-site motifs for the human NimA-related kinases. (**A**) Dendrogram showing the phylogenetic relationship between all human kinases. The branch containing the Nek-family is magnified in the inset. Adapted from Manning et al. (2002). (**B**) Cartoon depicting the identified and suggested functions for the Nek kinases in interphase cells (top left panel) and cells in different stages of mitosis (other three panels). See introduction for details. (**C**) Phosphorylation-site motifs of the Nek-family members were determined by OPLS. Shown are representative OPLS dot blots, and sequence logos of the favored residues quantified over multiple assays (n ≥ 2).

DOI: https://doi.org/10.7554/eLife.44635.002

The following figure supplements are available for figure 1:

**Figure supplement 1.** OPLS experimental design and Nek kinase isolation.

DOI: https://doi.org/10.7554/eLife.44635.003

**Figure supplement 2.** OPLS-results for kinase-dead Nek mutants.

DOI: https://doi.org/10.7554/eLife.44635.004

**Figure supplement 3.** Phosphorylation-site motifs for the Nek kinases and Plk1.

DOI: https://doi.org/10.7554/eLife.44635.005

**Figure supplement 4.** Nek2 preferentially phosphorylates a serine, rather than a threonine, phosphoacceptor residue.

*Figure 1 continued on next page*

Figure 1 continued

DOI: https://doi.org/10.7554/eLife.44635.006

threonine at the phospho-acceptor site, and one other fixed amino acid on a position ranging from five residues upstream (−5 position) to four residues downstream (+4 position) of the phospho-acceptor residue. The fixed amino acid is one of the 20 naturally occurring amino acids, or either phospho-threonine (pT) or phospho-tyrosine (pY). The remaining eight unfixed positions in the library contain an equimolar mix of all natural amino acids except cysteine, serine or threonine. The preference of the protein kinase for each residue at each position is then determined by directly comparing its activity against each individual peptide library. Lastly, to study phospho-acceptor site specificity of the kinase, we included three completely degenerate peptide libraries that contain either a fixed serine, threonine or tyrosine phospho-acceptor site, but no cysteine, serine, threonine or tyrosine in the degenerate positions.

Each of the Nek kinases were expressed as 3xFLAG-tagged variants in HEK 293T cells, isolated by anti-FLAG immunoprecipitation (IP; *Figure 1—figure supplement 1B*), and assayed by OPLS. For Nek1, Nek3, Nek6, Nek7, Nek8, and Nek9 the full-length protein was used. For Nek4 and Nek5, the kinase activity of the full-length molecule was low, and OPLS experiments were therefore performed using the isolated kinase domains, which displayed more activity. For Nek11, neither the full-length construct nor the isolated kinase domain displayed sufficient activity for motif determination, but clear motif data was obtained for all the other Nek-kinases (*Figure 1C*; Figure 4A,B). To ensure that the phosphorylating activity assayed by OPLS directly corresponded to that of the FLAG-tagged Nek kinase of interest, inactivating point mutants of each Nek kinase were generated, expressed in HEK 293T cells, purified and assayed as above by OPLS. As shown in *Figure 1—figure supplement 2*, none of the kinase-dead (KD) point mutants showed any detectable phosphorylation of the peptide libraries, demonstrating that the signal obtained with the wild-type Nek kinases was generated by the purified Neks and not by a contaminating kinase (*Figure 1—figure supplement 2*). Each OPLS-experiment was performed at least twice, and the data from all replicates were pooled to obtain phosphorylation-site motifs, which are presented as logos showing the positively selected residues in *Figure 1C*, and as complete logos including the negatively selected residues in *Figure 1—figure supplement 3*.

These OPLS experiments revealed that all of the tested Nek kinases display a clear preference for leucine, methionine, phenylalanine or tryptophan at the −3 position (*Figure 1C*). This is consistent with what has been reported for Nek6, and similar to what we found previously for Nek2 (*Lizcano et al., 2002*; *Alexander et al., 2011*). In addition, all the Nek kinases strongly selected against a proline in the + 1 position (*Figure 1C*; *Figure 1—figure supplement 3*). However, the OPLS-experiments also demonstrated clear specificity differences between the different family-members. Most notably, our results revealed a dichotomy in phospho-acceptor site preference within the family, with Nek1/3/4/5/8 preferentially phosphorylating threonine residues, while Nek6/7/9 primarily targeted peptides with serine residues as the phospho-acceptor (*Figure 1C*). Since the phosphorylation-site motif determined previously for Nek2 did not contain information on acceptor site specificity, an *in vitro* kinase assay of isolated Nek2 with degenerate peptide libraries containing either a serine or threonine phospho-acceptor site was performed. Nek2 phosphorylated the serine phospho-acceptor library more efficiently than the threonine library, indicating that Nek2 shares a preference for a phospho-acceptor serine residue with Nek6/7/9 (*Figure 1—figure supplement 4*). Together, these data reveal a Nek kinase family core consensus motif of [LMFW]-X-X-S/T-[no P], but with additional specificity-differences. In general, Nek1/3/4/5/8 displayed a preference for hydrophobic and basic residues on multiple positions within the motif, while Nek6/7/9 displayed a preference for acidic residues and selected hydrophobic residues at various positions. The separation between these groups of Nek kinases based on substrate-specificity is discussed in more detail below.

## Validation of the phosphorylation-site motif for all members of the Nek family

To validate our OPLS-results, we designed peptides predicted by our OPLS-results to be good substrates for either the Nek1/3/4/5/8 (13458-tide) group of kinases, or the Nek6/7/9 (679-tide) group of kinases. In addition, we synthesized variants of these peptides that contained residues predicted to be disfavored at specified positions (*Figure 2A*). We tested the phosphorylation of these peptides using *in vitro* kinase assays with Nek3 and Nek7 as representative members of these Nek kinase groups (*Figure 2B–D*). As shown in *Figure 2B,C*, Nek3 could readily phosphorylate the optimal Nek13458-tide but was substantially impaired in its ability to phosphorylate peptides in which either the −3 or −2 hydrophobic residues were mutated to aspartic acid (*Figure 2B,C*). As shown in *Figure 2D*, Nek7 phosphorylated the optimal 679-tide much more rapidly than a 679-tide variant containing a disfavored hydrophobic residue (I) in place of the favored hydrophobic residue (L) in the −3 position (*Figure 2D*). Similarly, substitution of the favored aspartic acid residue in the −2 position for a basic residue (R), substantially reduced peptide phosphorylation by Nek7 (*Figure 2D*).

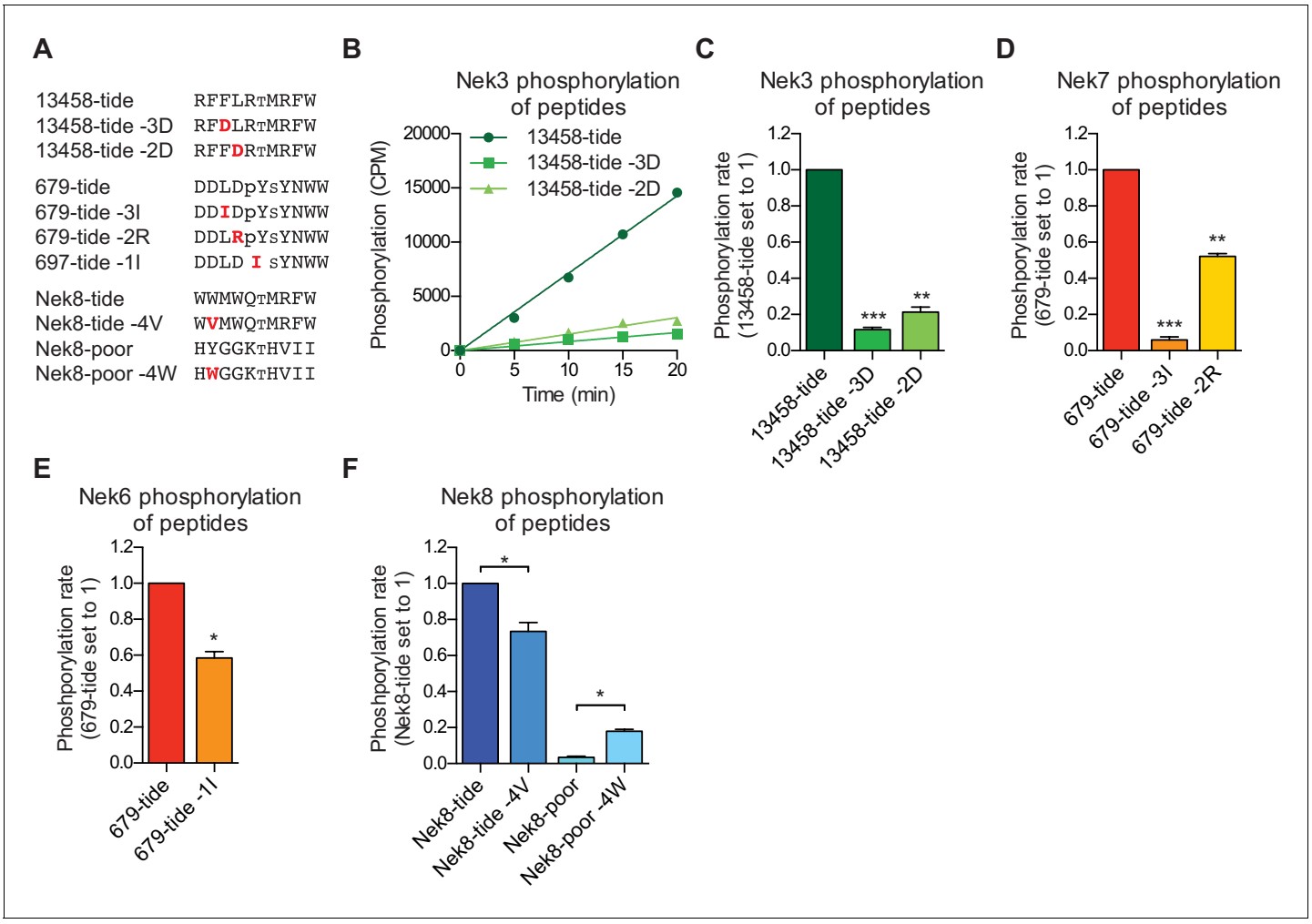

**Figure 2.** Validation of the phosphorylation-site motifs for the Nek kinases. (**A**) Peptides used in the studies depicted in panels B-F. Small font S or T indicates the phospho-acceptor site, pY indicates a phosphorylated tyrosine residue. (**B**) Purified Nek3 was incubated with the indicated peptide substrates and radiolabeled ATP. Peptide phosphorylation was quantified by scintillation counting (Mean ± SEM, n = 3; CPM = counts per minute). (**C**) The data depicted in panel (B) were analyzed by linear regression and the slopes of the regression curves were calculated to determine the phosphorylation rate (Mean ± SEM, n = 3, **p<0.005, ***p<0.0005). (**D**) As in panel (C), but for Nek7 (Mean ± SEM, n = 3, **p<0.005, ***p<0.0005). (**E**) As in panel (C), but for Nek6 (Mean ± SEM, n = 3, *p<0.05). (**F**) As in panel (C), but for Nek8 (Mean ± SEM, n = 3, *p<0.05).
DOI: https://doi.org/10.7554/eLife.44635.009

Interestingly, a subset of Nek kinases, including Nek7, Nek8 and particularly Nek6, displayed unexpected selection for phospho-tyrosine in the −1 position, suggesting that these kinases might target substrates that have been primed by tyrosine kinase mediated phosphorylation. To validate this specificity, we substituted the phospho-tyrosine in the −1 position of 679-tide to an OPLS-disfavored isoleucine residue, which resulted in a reduced rate at which Nek6 phosphorylated 679-tide (*Figure 2E*). Finally, all the Nek kinases displayed a strong preference for a tryptophan in the −4 position. Selection for tryptophan within kinase substrate motifs, especially at the −4 position, has rarely been observed (*Miller and Turk, 2018*). Corroborating our OPLS-data, the rate at which Nek8 phosphorylated its optimal substrate Nek8-tide was reduced upon substitution of the tryptophan in the −4 position to a valine (*Figure 2F*). Furthermore, introduction of a tryptophan in the −4 position in a peptide that is otherwise a very poor substrate for Nek8 (Nek8-poor) significantly enhanced the rate at which Nek8 could phosphorylate Nek8-poor (*Figure 2F*). Taken together, these results with optimal and suboptimal peptide variants validate our OPLS-data, and demonstrate that we can use the OPLS-data to predict good and poor substrates for the Nek kinases.

## Nek10 is a dual-specificity kinase

Curiously, in these OPLS experiments, Nek10 displayed very little sequence preference for any residue, except for a tyrosine at every position (*Figure 3A*). We hypothesized that this result could indicate that Nek10 is a tyrosine kinase. This could explain the effective phosphorylation of all peptide libraries containing a fixed tyrosine, because the tyrosine itself could serve as a phospho-acceptor site. In addition, it would explain the absence of additional preferences, because each of the 198 peptide libraries will contain tyrosine residues on non-fixed positions and therefore have many different target sites (*Figure 1—figure supplement 1A*). In agreement with this hypothesis, a preliminary, non-comprehensive mass spectrometric experiment to analyse phospho-sites on *in vitro* auto-phosphorylated Nek10 revealed the presence of several phosphorylated tyrosine residues (*Figure 3—figure supplement 1A,B*). To directly determine if Nek10 phosphorylates tyrosine residues, we designed a Nek10 substrate peptide based on the auto-phosphorylation site data, and introduced either a tyrosine, serine, or threonine at the phospho-acceptor site (*Figure 3B,C*). Surprisingly, Nek10 effectively phosphorylated peptides with a serine phospho-acceptor site, but even more rapidly phosphorylated peptides with a tyrosine phospho-acceptor site (*Figure 3B,C*). In contrast, substitution of threonine as the phospho-acceptor significantly impaired peptide phosphorylation by Nek10 (*Figure 3B,C*). We therefore conclude that Nek10 is a tyrosine/serine dual-specificity kinase.

None of the other Nek kinases phosphorylated peptides libraries containing tyrosine as the phospho-acceptor (0 position in *Figure 1C*, also *Figure 3—figure supplement 2A*). Therefore, we reasoned that sequence comparison between the Nek kinases might reveal residues that permit Nek10 to phosphorylate both serine and tyrosine residues. Structural analysis has implicated specific regions of the kinase domain that contribute to serine/threonine or tyrosine specificity (*Taylor et al., 1995*), and one of these regions is the P+1 loop, which lies directly N-terminal of the conserved APE-motif (*Figure 3C*). We therefore aligned the P+1 loop sequences of the Nek kinase family members, and noted that Nek10 has an isoleucine whereas all the other Nek kinases have a proline at the position four residues N-terminal of the APE-motif (APE-4; *Figure 3E*). Interestingly, an isoleucine in the APE-4 is typically found in tyrosine kinases, but in less than 5% of serine/threonine kinase, which more commonly have a proline on this position (*Figure 3F*). We therefore mutated the APE-4 isoleucine in Nek10 to a proline (I693P) and found that this increased its capacity to phosphorylate a substrate peptide on a serine residue but rendered Nek10 incapable of phosphorylating the same peptide on a tyrosine (*Figure 3G*).

Another region contributing to phospho-acceptor site specificity is the catalytic loop, which contains the aspartic acid which serves as the catalytic base within the conserved HRD motif (*Figure 3D*; *Taylor et al., 1995*). Intriguingly, Nek10 has a threonine in the HRD+2 position, which is uncommon within the serine/threonine kinome. Instead, 86% of serine/threonine kinases contain a lysine residue in the HRD+2 position, including all the other Nek kinases (*Figure 3H*; *Figure 3—figure supplement 2B*). Mutation of the HRD+2 threonine to a lysine (T657K) generated a Nek10 variant that, like Nek10 I693P, could effectively phosphorylate serine residues but was incapable of phosphorylating tyrosine residues (*Figure 3I*). Finally, we sought to make a variant version of Nek10 that would exclusively phosphorylate substrates on tyrosine residues. To this end, we mutated the threonine in the APE-5 position to a proline, which is highly conserved in tyrosine kinases, but present in only three

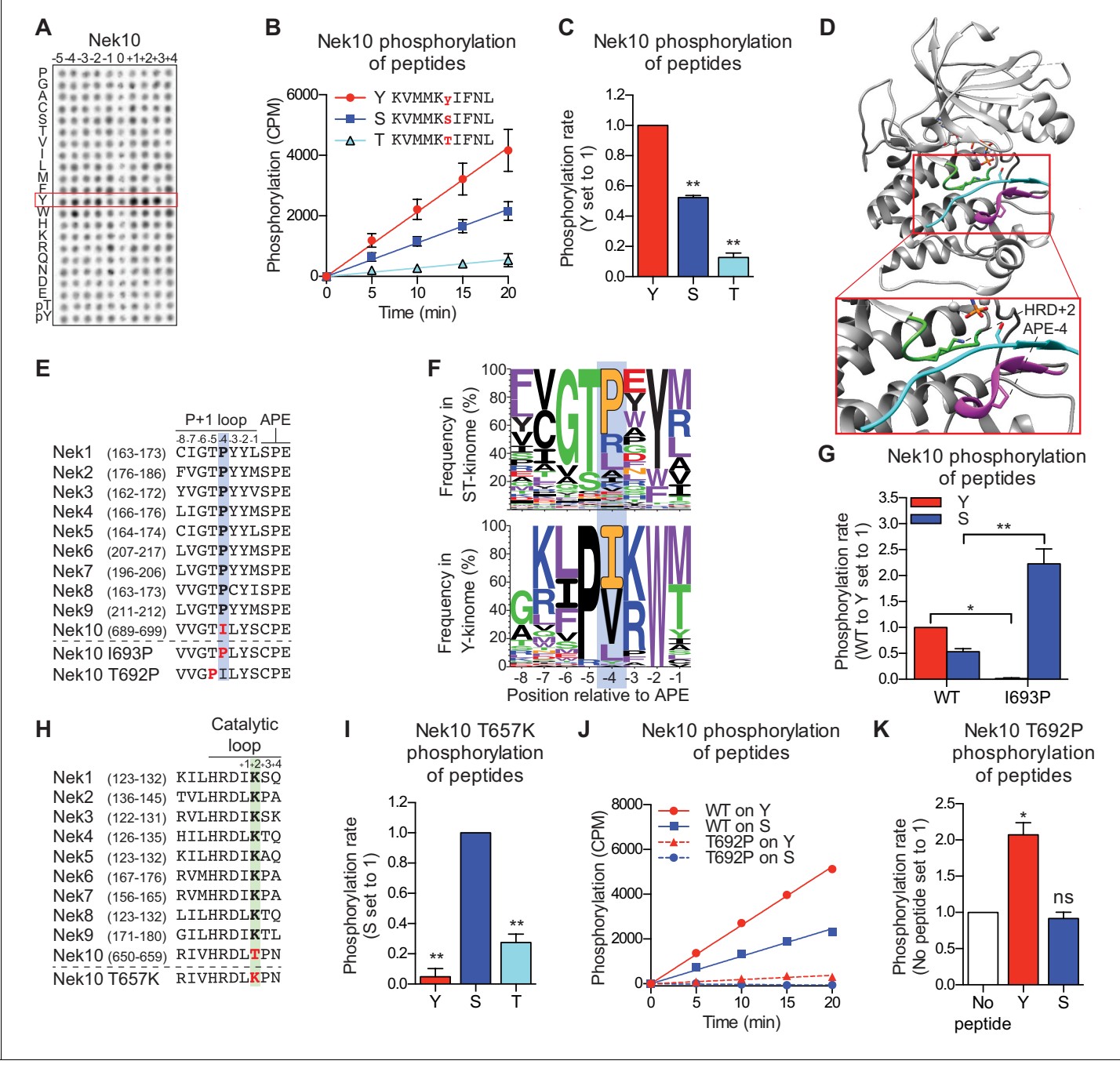

**Figure 3.** Nek10 is a dual-specificity kinase. (**A**) Representative blot showing the OPLS results for wild-type Nek10 on the serine-threonine peptide library. (**B**) Purified Nek10 was incubated with the indicated peptide substrates and radio-labeled ATP. Peptide phosphorylation was quantified by scintillation counting. (Mean ± SEM, n = 3). (**C**) The data depicted in panel (**B**) were analyzed by linear regression and the slopes of the regression curves were calculated to determine the phosphorylation rate (Mean ± SEM, n = 3, **p<0.005). (**D**) Cartoon representation of the Akt/GSK3-peptide structure (PDB: 1O6L). The kinase domain, activation loop, and the P+1 loop are shown in gray, green and magenta, respectively. The GSK3 peptide is shown in cyan, side chain shown is of the phospho-acceptor serine. (**E**) Amino acid sequence alignment of the P+1 loop and APE motif of Nek1 to Nek10, and including the Nek10 I693P and T692P variants. (**F**) Logo depicting the frequency of amino acids in the P+1 loop of either all human serine/threonine (ST) kinases, but excluding the tyrosine kinase like group (upper panel), or all human tyrosine (Y) kinases (lower panel). (**G**) As in panel (**C**), but for Nek10 WT and the I693P variant (Mean ± SEM, n = 3, *p<0.05, **p<0.005). (**H**) Amino acid sequence alignment of the catalytic loop of Nek1 to Nek10, including the Nek10 T657K variant. (**I**) As in panel (**C**), but for the Nek10 T657K variant (Mean ± SEM, n = 3, **p<0.005). (**J**) As in panel (**B**), but for Nek10 WT and the T692P variant. (**K**) As in panel (**C**) but for the Nek10 T692P variant. In order to obtain accurate rate calculations, the experiment

*Figure 3 continued on next page*

*Figure 3 continued*

shown in panel J was repeated x3 with longer incubation times (Mean ± SEM, n = 4, *p<0.05). The signal observed in the 'no peptide' sample results from binding of auto-phosphorylated kinase to the phosphocellulose paper.

DOI: https://doi.org/10.7554/eLife.44635.010

The following figure supplements are available for figure 3:

**Figure supplement 1.** Nek10 autophosphorylates on serine, threonine, and tyrosine *in vitro*.

DOI: https://doi.org/10.7554/eLife.44635.011

**Figure supplement 2.** The ability of Nek10 to phosphorylate on tyrosine is unique within the Nek family.

DOI: https://doi.org/10.7554/eLife.44635.012

**Figure supplement 3.** An APE-4 isoleucine in Nek1 enhances substrate phosphorylation on tyrosine residues.

DOI: https://doi.org/10.7554/eLife.44635.013

serine/threonine kinases (*Figure 3F*). In contrast to the I693P and T657K mutations, this APE-5 mutation (T692P) severely reduced overall Nek10 kinase activity (*Figure 3J*). However, the resulting Nek10 T692P variant could still detectably phosphorylate tyrosine residues while it was unable to phosphorylate serine residues (*Figure 3K*). Taken together, these results identify specific residues in the kinase domain of Nek10 that are essential for its dual-specificity, and show that point-mutations in the catalytic or P+1 loop can toggle Nek10 between being a dual-specificity, serine-specific, or tyrosine-specific kinase.

## An APE-4 isoleucine in Nek1 enhances substrate phosphorylation on tyrosine residues

We next investigated whether introduction of an HRD+2 threonine and/or APE-4 isoleucine in a serine/threonine kinase would be sufficient to induce tyrosine-directed activity. These mutational experiments were conducted using Nek1, since its structure is known (*Melo-Hanchuk et al., 2017*), wild-type Nek1 did not efficiently phosphorylate peptide substrates on tyrosine in our experiments (*Figure 3—figure supplement 2A*), and, like all other Nek kinases except Nek10, Nek1 has an HRD +2 lysine and APE-4 proline (*Figure 3E,H*). However, murine Nek1 has been reported to be a dual-specificity kinase (*Letwin et al., 1992*), suggesting that Nek1 might have intrinsic tyrosine-phosphorylation capacity in specific contexts, and might therefore be more susceptible than other serine/threonine kinases to mutations that broaden phospho-acceptor specificity.

Mutants of Nek1 containing a threonine in the HRD+2 position (Nek1 HRD+2), isoleucine on the APE-4 position (Nek1 APE-4), or both (Nek1 Double Mutant; DM), were generated and the recombinant kinases were tested for their capacity to phosphorylate peptide substrates containing either a threonine or tyrosine phospho-acceptor site (*Figure 3—figure supplement 3A,B*). Notably, either point mutation reduced kinase activity ~ 6-fold to 8-fold compared to Nek1 WT, while the combination of mutations almost completely abrogated Nek1 activity (*Figure 3—figure supplement 3B*). To facilitate comparison of threonine and tyrosine-directed activity for all the Nek1 variants, these experiments were therefore repeated using a 6-fold reduced concentration of Nek1 WT compared to the mutants (*Figure 3—figure supplement 3C,D*). As seen in *Figure 3—figure supplement 3C,D* Nek1 WT and both mutants phosphorylated peptides on threonine residues better than on tyrosine residues (note the differences in the Y-axis scale), Nonetheless, the APE-4 isoleucine mutant, but not the HRD+2 threonine mutant, enhanced the relative preference of Nek1 for tyrosine substrates (*Figure 3—figure supplement 3E*) to ~ 20% of the efficiency with which it phosphorylated threonine-containing peptides. The same was observed for the Nek1 DM variant, although this result should be interpreted cautiously, given the low activity of Nek1 DM (*Figure 3—figure supplement 3E*). Nek10, however, in contrast to the Nek1 APE-4 isoleucine mutant, phosphorylates tyrosine substrates with at least twofold greater activity than the same peptide on a serine or threonine phosphoacceptor (*Figure 3C*), indicating that additional sequence determinants in the Nek10 kinase domain must collaborate with the APE-4 isoleucine residue to enhance dual-specificity.

## Nek10 phosphorylates serines and tyrosines in different sequence contexts

The positioning of the substrate in the catalytic site of the kinase is different for serine/threonine kinases compared to tyrosine kinases (*Taylor et al., 1995*), raising the question of how a dual-specificity kinase is able to correctly position substrates to catalyze phosphorylation on serine as well as tyrosine residues. We reasoned that the phosphorylation-site sequence context might play a major role, and therefore set out to determine whether Nek10 recognizes distinct phosphorylation-site motifs for serine and tyrosine substrates. In these experiments we used a serine-specific variant of Nek10 (I693P) to eliminate issues arising from tyrosine-phosphorylation of peptide libraries by WT Nek10 that made the previous OPLS-results uninterpretable (*Figure 3A*). As shown in *Figure 4A*, for Nek10 I693P the strong preference for tyrosine in every position was absent, but instead it displayed a clear preference for hydrophobic and basic residues, similar to the Nek1/3/4/5/8 group (*Figures 1C*, *4A*). However, in contrast to other members of this group that preferentially select for threonine as the phospho-acceptor, Nek10 I693P preferentially phosphorylates serine (*Figure 4A*). To eliminate the possibility that the I693P mutation affected the intrinsic serine-motif specificity of Nek10, we validated these OPLS-results with WT Nek10, rather than with the Nek10 I693P mutant. An optimal serine peptide substrate ((S)-tide) was synthesized on basis of the OPLS-results, along with variants in which the favored hydrophobic residue in the −3 position and the favored basic residue in the −1 positions were replaced by a disfavored aspartic acid residue (−3D, −1D), and these peptides were used as substrates in an in vitro kinase assays. As shown in *Figure 4B*, WT Nek10 could effectively phosphorylate (S)-tide but not the −3D and −1D peptides, therefore corroborating our OPLS-results (*Figure 4B*, *Figure 4—figure supplement 1*).

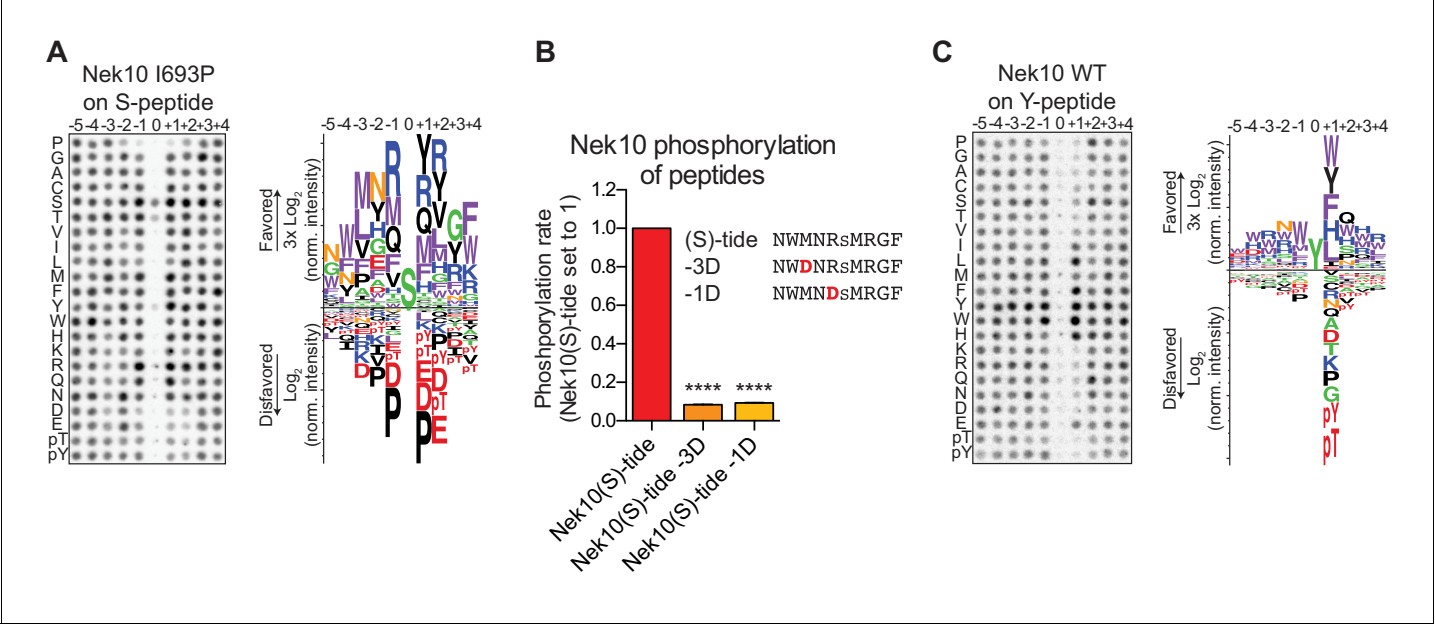

**Figure 4.** Phosphorylation-site motifs for Nek10 on tyrosine and serine target sites. (A) The phosphorylation-site motif of Nek10 I693P was determined by OPLS. Shown are a representative blot, and a sequence logo of data quantified over two independent OPLS-experiments, showing both the favored and disfavored amino acids. (B) Purified Nek10 was incubated with the indicated peptide substrates and radiolabeled ATP. Peptide phosphorylation was quantified by scintillation counting. The phosphorylation rate was determined by linear regression (Mean ± SEM, n = 3, ****p<0.00005). (C) As in panel (A), but now for WT Nek10, and using a peptide library with a fixed tyrosine as the phosphoacceptor site (n = 3). Note that in this case the 0 position column contains no peptide libraries.

DOI: https://doi.org/10.7554/eLife.44635.007

The following figure supplement is available for figure 4:

**Figure supplement 1.** Validation of the Nek10 phosphorylation-site motif on serine-substrates.

DOI: https://doi.org/10.7554/eLife.44635.008

To investigate the phosphorylation-site motif for Nek10 on tyrosine substrates, we designed a peptide library identical to the serine/threonine peptide library, but with a tyrosine instead of a serine/threonine mixture in the phospho-acceptor site. Importantly, to prevent phosphorylation of the degenerate positions, tyrosine, together with serine, threonine and cysteine residues, was excluded from the amino acid mixture in the degenerate positions. These OPLS experiments revealed a very different Nek10 phosphorylation-site motif for tyrosine substrates compared to serine substrates. On tyrosine substrates, Nek10 does not select for the canonical Nek-family motif signature [LMFW]-X-X-S/T-[no P], but instead shows a near-absolute requirement for aromatic or selected hydrophobic residues in the +1 position (*Figure 4C*). Thus, the serine phosphorylation-site motif of the dual-specificity kinase Nek10 is similar to that of the other Nek kinases, but the tyrosine phosphorylation-site motif of Nek10 is substantially more restricted.

## Nek10 auto-phosphorylates on serine and tyrosine residues *in vitro* and in cells

Next, we investigated whether Nek10 can phosphorylate protein substrates, in addition to peptide substrates, on both serine and tyrosine residues. Unfortunately, no Nek10 substrates have been reported to date. However, preliminary mass spectrometry data indicated that Nek10 is a substrate for auto-phosphorylation (*Figure 3—figure supplement 1A,B*). Indeed, incubation of Nek10 with radiolabeled ATP revealed efficient auto-phosphorylation of WT and serine-specific Nek10 (I693P (Ser)), but not of kinase-dead (KD) Nek10 in which the catalytic aspartic acid was mutated to an asparagine residue (*Figure 5A*). Furthermore, despite its reduced activity, we could also clearly detect auto-phosphorylation of the tyrosine-specific (T692P(Tyr)) Nek10 variant, demonstrating that Nek10 can undergo auto-phosphorylation on both serine and tyrosine residues in vitro (*Figure 5A*).

Of the 518 protein kinases only 18 are recognized as dual-specificity. Members of the GSK3 and DYRK serine/threonine kinase families can also auto-phosphorylate on tyrosine residues, but do this exclusively (GSK3) or primarily (DYRK) during folding *in cis* (*Cole et al., 2004*; *Soundararajan et al., 2013*). To test whether Nek10 can auto-phosphorylate on tyrosine and serine *in trans*, we performed *in vitro* kinase assays using FLAG-tagged WT, serine-specific or tyrosine-specific Nek10 to phosphorylate an HA-tagged KD Nek10 as substrate. Following incubation, Nek10 kinase and substrate were separated by anti-FLAG and anti-HA IP, respectively, and analyzed by immunoblotting and phosphorimaging. As shown in *Figure 5B*, phosphorylation of KD Nek10 was observed following incubation with either a serine-specific or tyrosine-specific kinase-proficient Nek10, demonstrating that *in vitro*, Nek10 can auto-phosphorylate *in trans* on both serine and tyrosine residues (*Figure 5B*).

Next, to investigate if serine and tyrosine auto-phosphorylation of Nek10 could also occur in cells, we generated Nek10 knock-out U-2 OS cell lines using CRISPR technology, and validated the knock-outs using genomic sequencing of the target site (*Figure 5—figure supplement 1A*), and with TIDE analysis (*Brinkman et al., 2014*). These were then reconstituted with doxycycline-inducible, 3xFLAG-tagged WT, KD, serine-specific (I693P(Ser)) and tyrosine-specific (T692P(Tyr)) variants of Nek10 (*Figure 5C*). Tyrosine phosphorylation was assessed by anti-FLAG IP followed by immunoblotting with an antibody recognizing phospho-tyrosine. Clear tyrosine phosphorylation was observed of WT Nek10, but not of a KD or serine-specific variant, consistent with Nek10 auto-phosphorylation on tyrosine residues in cells (*Figure 5D*). In contrast to what we found *in vitro*, however, we could not detect any tyrosine phosphorylation on tyrosine-specific Nek10 (*Figure 5D*), likely due to a combination of the very low activity of this Nek10 variant, and the higher threshold required for antibody detection compared to $^{32}$[P]-radiolabeling.

To map the tyrosine(s) targeted for auto-phosphorylation by Nek10, we used the OPLS phosphorylation-site motif to select seven candidate Nek10 target sites, and mutated each of these sites to a phenylalanine. WT and mutant kinases containing 3xFLAG-tags were expressed in wild-type, non-edited U-2 OS cells, followed by anti-FLAG IP and immunoblotting for phospho-tyrosine. As shown in *Figure 5E*, tyrosine-phosphorylation was detected on each of the mutants, but the signal was substantially reduced upon mutation of Y644 (*Figure 5E*). A possible interpretation of these results is that Y644 is a major auto-phosphorylation target site of Nek10. To test this hypothesis, we performed an *in vitro* kinase assay using an active tyrosine-specific form of Nek10 to phosphorylate *in trans* either a KD Nek10 Y644 control, or a KD Nek10 containing the Y644F mutation (*Figure 5F*). This experiment confirmed that *in vitro*, Nek10 directly auto-phosphorylates Y644 *in trans* (*Figure 5F*). Interestingly, the Y644F mutation dramatically reduced the activity of Nek10 in an *in*

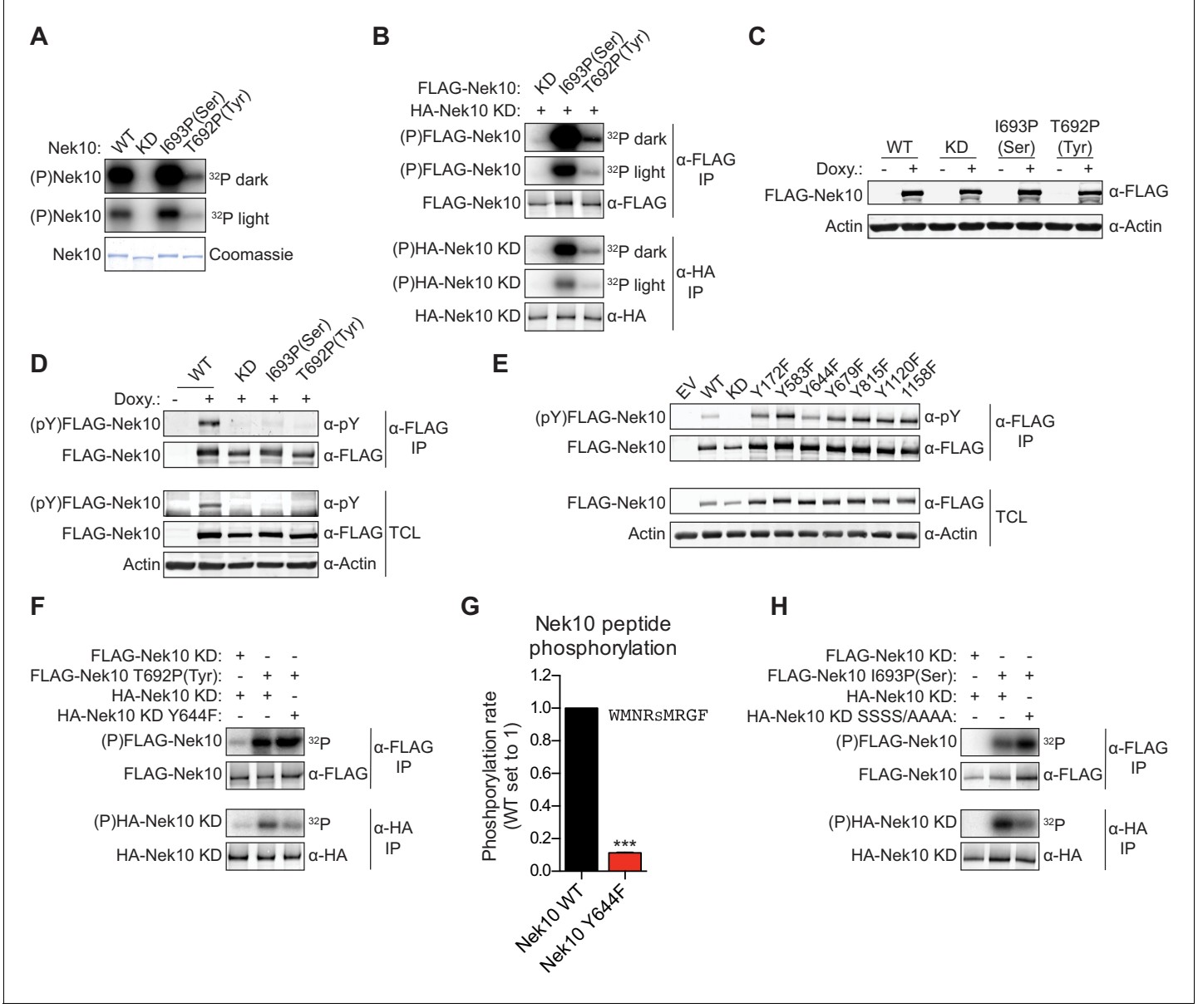

**Figure 5.** Nek10 autophosphorylates on both tyrosine and serine residues. (**A**) Purified wild-type (WT), kinase-dead (KD), serine-specific (I693P(Ser)), or tyrosine-specific (T692P(Tyr)) Nek10 was incubated *in vitro* with radiolabeled ATP. Total Nek10 levels were analyzed by coomassie-staining, while phosphorylated (P) Nek10 was detected by phosphorimaging. $^{32}$P dark and $^{32}$P light respectively indicate a longer and shorter exposure of the same gel. (**B**) *In vitro* kinase assay with indicated FLAG-tagged Nek10 variants as the kinase, and HA-tagged Nek10 KD as the substrate. Kinase-proficient and kinase-dead Nek10 were separated by anti-FLAG and anti-HA IP respectively, and analyzed by immunoblotting and phosphorimaging. (**C**) Nek10 KO U-2 OS cells were reconstituted with doxycycline (Doxy) inducible FLAG-Nek10 variants and treated with 2 µg/ml doxycycline for 7 days, or left untreated. FLAG-Nek10 levels were analyzed by immunoblotting. (**D**) Cells described in panel (**C**) were treated with doxycycline for 2 days, and FLAG-Nek10 was isolated by α-FLAG IP, followed by immunoblotting to detect total Nek10 (α-FLAG) and tyrosine-phosphorylated Nek10 (α-pY) (TCL = total cell lysate). (**E**) Wild-type, non-edited U-2 OS cells were transfected with an empty vector control (EV) or the indicated FLAG-Nek10 variants. For WT and KD Nek10, less plasmid was transfected than for the Nek10 Y-to-F mutants. FLAG-Nek10 was isolated by α-FLAG IP and analyzed by immunoblotting. (**F**) As in panel (**B**), now using FLAG-tagged Nek10, either KD or tyrosine-specific (T692P(Tyr)), as the kinase, and HA-tagged Nek10 KD, either lacking additional mutations or with a Y644F mutation, as the substrate. (**G**) Purified Nek10 WT or the Y644F mutant was incubated with a peptide substrate and radiolabeled ATP. Peptide phosphorylation was quantified by scintillation counting. The phosphorylation rate was determined by linear regression (Mean ± SEM, n = 3, ***p<0.0005). Note that the autophosphorylation-defective Y644F mutant has dramatically reduced activity. (**H**) As in panel (**B**), now using FLAG-tagged Nek10, either KD or serine-specific (I693P(Ser)), as the kinase, and HA-tagged Nek10 KD, either lacking additional mutations or with a S353A/S356A/S358A/S359A quadruple mutation (SSSS/AAAA), as the substrate.

DOI: https://doi.org/10.7554/eLife.44635.014

*Figure 5 continued on next page*

*Figure 5 continued*

The following figure supplement is available for figure 5:

**Figure supplement 1.** Nek10 autophosphorylates on Y644 and S358/359.
DOI: https://doi.org/10.7554/eLife.44635.015

*vitro* kinase assay with a serine peptide substrate (*Figure 5G*), and a similar result was obtained with a tyrosine peptide substrate (*Figure 5—figure supplement 1B*). These results strongly indicate that Nek10 auto-phosphorylation on Y644 contributes to its activity.

Finally, to study serine auto-phosphorylation in cells, we isolated WT, KD, serine-specific and tyrosine-specific Nek10 from the doxycycline-induced Nek10 knock-out cells (*Figure 5C*), and analyzed its phosphorylation status by mass spectrometry. Multiple phosphopeptides were detected, but the region corresponding to N346-R365 of Nek10, which is outside of the kinase domain, was the most abundant phosphopeptide in the WT and serine-specific samples, and was completely absent in the KD and tyrosine-specific samples (*Figure 5—figure supplement 1C,D*). This peptide contains six serine residues, of which four were identified as phosphorylated by MASCOT protein identification software (*Figure 5—figure supplement 1C*). Mutation of these four serine sites in KD Nek10 reduced its phosphorylation *in trans* by a serine-specific Nek10 variant, demonstrating that one or more of these sites can be auto-phosphorylated *in trans* by Nek10 (*Figure 5H*). To determine which of these four sites was the most likely target site, we manually annotated the MS/MS spectra, and identified S358 of Nek10 as the phosphosite most consistent with the mass spectrometry data (*Figure 5—figure supplement 1E*). Therefore, we conclude from these results that Nek10 can auto-phosphorylate both *in vitro* and in cells, that *in vitro* auto-phosphorylation can occur *in trans*, and we identified Y644 in the kinase domain, and S358 outside of the kinase domain, as auto-phosphorylation target sites.

## The Nek kinase family diverges into four specificity-groups

The OPLS results indicated that there are substantial differences in specificity among the Nek kinase family members (*Figure 1C*). To examine this in more detail, we performed hierarchical clustering of the Nek kinome based on quantitative OPLS motif information. To also include Nek10 in this analysis, we used the serine phosphorylation-site OPLS motif (Nek10(S)), because the tyrosine phosphorylation-site motif is very distinct from the motifs of the other Nek kinases. This clustering revealed that Nek1/3/4/5/8 were clearly separated from Nek2/6/7/9/10(S) (*Figure 6A*). Importantly, this division of the Nek kinases based on preferred substrate sequence cannot be explained by amino acid sequence similarity between the Nek kinase domains themselves (*Figure 6B*). Instead, an important contributor to this hierarchy is the preference for either serine or threonine as the phospho-acceptor site (*Figure 6C*). To assess non-phosphosite contributions, we re-clustered the Nek kinase family based on motif specificity, but excluded the phospho-acceptor site preference (*Figure 6D*). This revealed that within the group of threonine-directed Nek kinases, Nek 1/3/4 cluster separately from Nek 5/8 (*Figure 6D*). Overall then, the motif data indicate that the Nek kinase family can be separated into four specificity groups based on phospho-acceptor preference and additional amino acids specificities (*Figure 6D*). Interpretation of the sequence logos shown in *Figure 1C* indicates that Nek 1, 3 and 4 cluster in Group 1 based on their shared preference for arginine in the −1 position. Within Group 1, Nek 1 and 4 also show strong selectivity for lysine and arginine in the +2 position, and they are exceptional in their lack of preference for a tryptophan in the −3 position. Group 2 consists of Nek 5 and 8, and is closely related to Group 1, but has a less prominent selection for arginine in the −1 position. Group 3, which consists of Nek 2 and 10, are distinguished by their serine phospho-acceptor specificity (*Figure 6C*; *Figure 3—figure supplement 2A*), although the remaining motif specificity of Nek two is similar to that of kinases in Group 2 (*Figure 1—figure supplement 3*). Finally, Group 4 consists of Nek 6, 7, and 9, and clusters distantly from the rest (*Figure 6D*) based on a preference for acidic residues in the −5,−4, and particularly in the −2 position that is not shared with any of the other Nek kinases (*Figure 6E*; *Figure 6—figure supplement 1*).

To rationalize the unusual selectivity for acidic residues displayed by Group 4 Nek kinases, we compared the X-ray crystal structures of the Nek1 and Nek7 kinase domains (*Haq et al., 2015*; *Melo-Hanchuk et al., 2017*). This revealed the presence of two basic regions in Nek7 that are

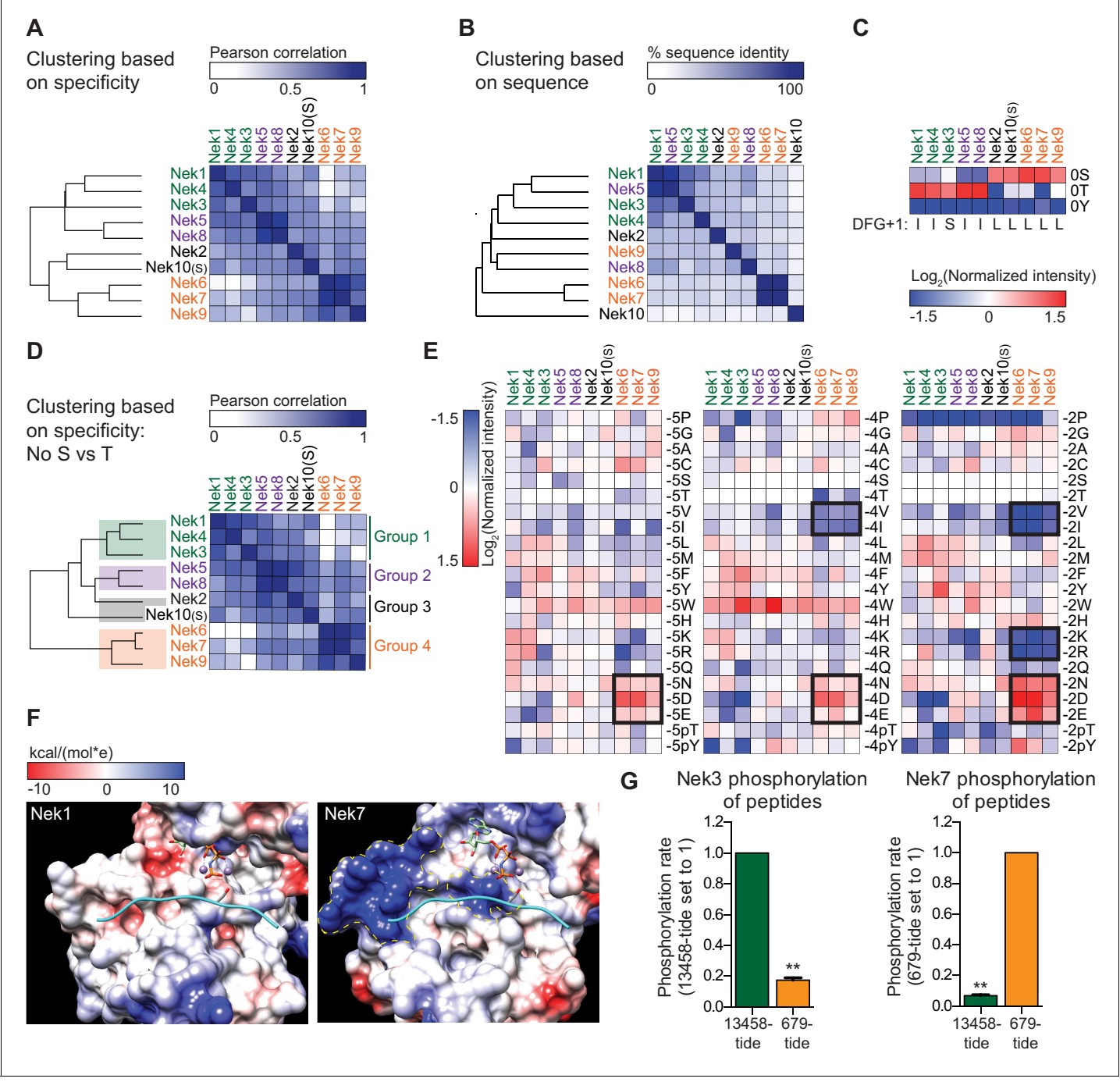

**Figure 6.** The Nek family diverges into four specificity-groups. (**A**) Similarity matrix and dendrogram depicting the Pearson correlation between the phosphorylation-site motifs of the Nek kinase family members. (**B**) Similarity matrix and dendrogram showing the percentage identity between the amino acid sequences of the Nek kinase domains. (**C**) Heatmap depicting the OPLS data for the phospho-acceptor site position. Measured intensities of individual dots on OPLS dot blots were normalized to the average of all values in the position (e.g. −1 position), averaged over multiple experiments, and log2-transformed (n ≥ 2). Nek family members are ordered according to the clustering shown in panel (**A**). For each Nek the amino acid C-terminal to the DFG-motif (DFG+1) is depicted underneath the heatmap. (**D**) As in panel (**A**), but the phospho-acceptor specificity data was excluded from the clustering analysis. (**E**) As in panel (**C**), but for the −5 (left panel), −4 (middle panel) and −2 (right panel) position to emphasize the distinct specificities of the Nek6/7/9 group (boxed sections). (**F**) Surface representation of the substrate binding sites of Nek1 (PDB: 4APC) and Nek7 (PDB:5DE2). Surface is shaded according to electrostatic potential. The dotted yellow lines indicate specific basic patches in the Nek7 structure that potentially interact with acidic residues in the substrate. Peptide and ANP-Mn was modeled into the Nek1 and Nek7 structures by alignment with the Akt/GSK3-peptide structure (PDB: 1O6L; **Figure 3D**). (**G**) Purified Nek3 (left panel) or Nek7 (right panel) was incubated with peptide substrates and

*Figure 6 continued on next page*

*Figure 6 continued*

radiolabeled ATP. Peptide phosphorylation was quantified by scintillation counting. The phosphorylation rate was determined by linear regression (Mean ± SEM, n = 3, **p<0.005).

DOI: https://doi.org/10.7554/eLife.44635.016

The following figure supplement is available for figure 6:

**Figure supplement 1.** Comparison of Nek-family phosphorylation-site motifs.

DOI: https://doi.org/10.7554/eLife.44635.017

absent from Nek1, and might accommodate acidic residues N-terminal of the phospho-acceptor site (*Figure 6F*). Additional crystal structures of these kinases with substrate peptides will be necessary to experimentally validate these observations. To further investigate the relevance of these motif differences between Group 4 and the other Nek kinases in substrate phosphorylation, we then performed *in vitro* kinase assays using Nek3 and Nek7 with either a good peptide-substrate for Groups 1 and 2 (13458-tide), or a good peptide-substrate for Group 4 (679-tide). As predicted, 13458-tide was a considerably better substrate for Nek3 than 679-tide, and this was reversed for Nek7 (*Figure 6G*).

## Nek6/7/9 share a common substrate motif with Polo-like kinase 1 and can phosphorylate the same substrates

We noticed a striking resemblance between the Nek6/7/9 motif, and the phosphorylation-site motif previously obtained for Plk1 (*Figure 7A*; *Alexander et al., 2011*). This resemblance is unlikely to result from Plk1 contamination of the Nek6/7/9 preparations since no phosphorylation signal was observed in OPLS-experiments performed using kinase-dead variants of these Nek kinases (*Figure 1—figure supplement 2*), unless the activity of contaminating Plk1 was dependent in some way on the activity of Nek6/7/9. To further address this potential concern, we analyzed the isolated Nek kinase preparations for co-purified Plk1 by immunoblotting (*Figure 7—figure supplement 1*). We could readily detect Plk1 in HEK 293T cell lysates, but could not detect Plk1 in any of the isolated Nek kinase preparations. We therefore conclude that the OPLS experiments identified the true substrate motifs for Nek6/7/9, and not of co-purified Plk1. We further analyzed the resemblance between the Nek6/7/9 and Plk1 motifs by hierarchical clustering of the Nek-family and Plk1 based on substrate motif data from OPLS. This revealed that, with regard to substrate specificity, Nek6/7/9 are more related to Plk1 than they are to the other Nek kinases (*Figure 7B*). Importantly Plk1, together with Cdk1, has previously been reported as the upstream activator of Nek6/7/9 (*Bertran et al., 2011*). Hence, Plk1 and Nek6/7/9 not only operate together in a linear signaling cascade during mitosis, but also share the same phosphorylation-site motif, and hence could potentially phosphorylate a common set of substrates.

To assess potential substrate overlap, we used the quantitative specificity information derived from the OPLS experiments to predict substrates for Plk1 and Nek6/7/9 using the *Scansite* scoring algorithm (*Obenauer et al., 2003*). We restricted the scoring to sites that are present in the *Phophositeplus* database of previously reported phosphorylation sites in the human proteome, to exclude sites that are potentially not surface-accessible (*Hornbeck et al., 2015*). We considered the top 10% of best scoring sites as potential targets. This analysis revealed a substantial overlap in candidate substrates among the different kinases: over 50% of Plk1 target sites were also predicted to be a good target site for at least one of the Nek kinases, and a total of 2801 sites were predicted to be good target sites for all four kinases (*Figure 7C*).

Next, we considered the possibility that substrates previously assigned as direct Plk1 targets on the basis of inhibitor or knockdown studies might actually be either direct substrates of Nek6/7/9, or are shared direct substrates of both Plk1 and Nek6/7/9. To address this possibility, we analyzed the candidate Plk1 substrates at the mitotic spindle that were obtained in a phosphoproteomic study, during which either a small molecule inhibitor or an shRNA was used to block Plk1 function (*Santamaria et al., 2011*). *Scansite* scoring of these substrates showed a strong correlation between the Nek7-scores and Plk1-scores (*Figure 7D*). Using a cut-off for the top 25% of best-scoring sites, we identified 109 distinct sites on 77 different proteins as potential targets of both Plk1 and Nek7. This included S453 on Cdc27, and S170 on RacGAP1, both of which have been established as direct

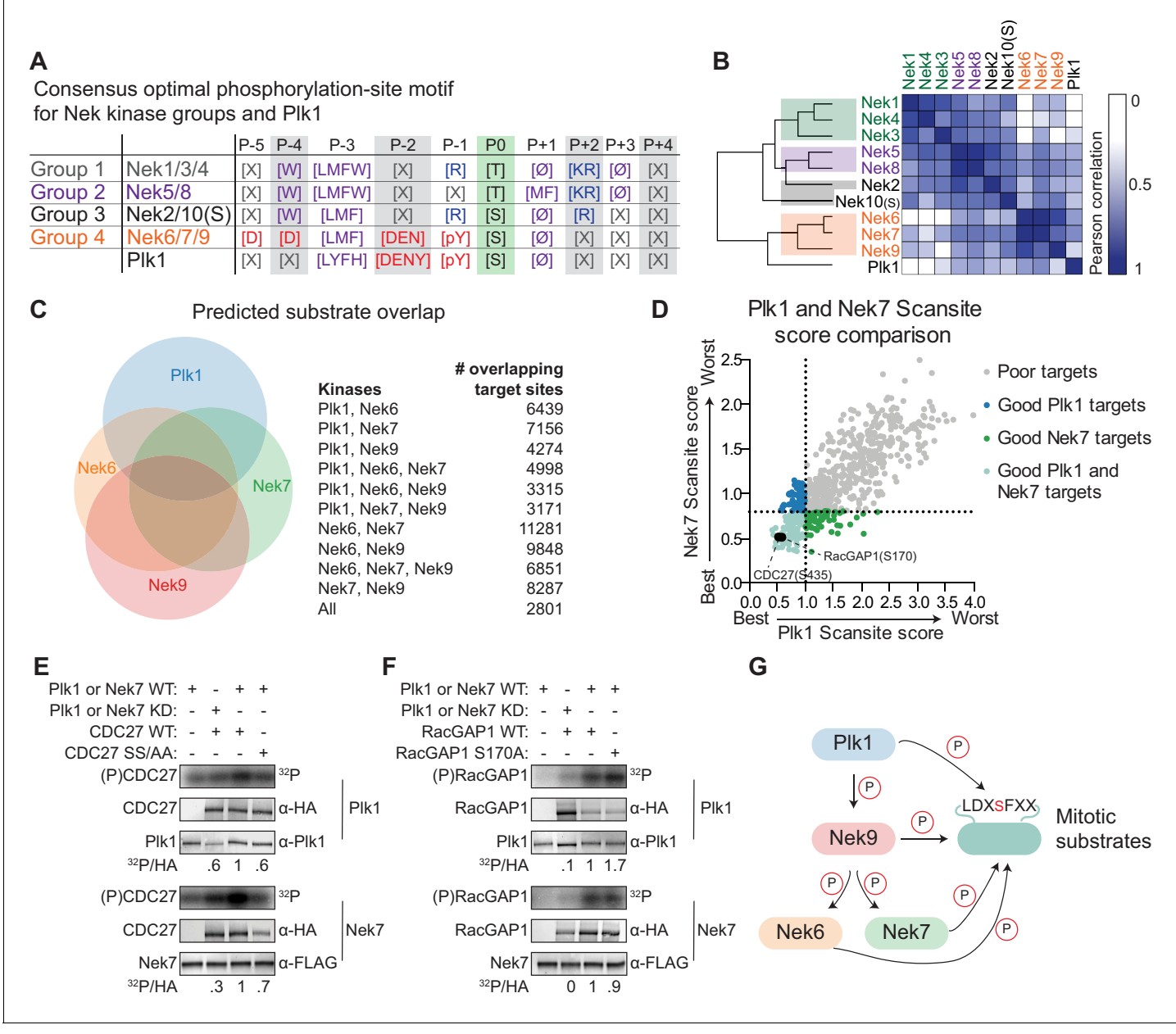

**Figure 7.** Plk1 and Nek6, 7 and 9 share a common phosphorylation-site motif. (**A**) Table depicting the consensus phosphorylation-site motifs for the different Nek specificity groups, and for Plk1, with Ø indicating hydrophobic amino acids. The Plk1 motif was generated from data published in *Alexander et al. (2011)*. (**B**) Similarity matrix and dendrogram depicting the Pearson correlation between the phosphorylation-site motifs of the Nek family members, and Plk1. (**C**) Sites in the *Phosphositeplus* database were scored using the *Scansite* algorithm for their match to the phosphorylation-site motifs of Plk1, Nek6, Nek7 and Nek9. The best scoring 10% were plotted in the Venn diagram to indicate the fraction of good potential target sites shared between the kinases. (**D**) Sites with reduced phosphorylation upon Plk1 inhibition or knockdown, as reported by *Santamaria et al. (2011)*, were scored for their match to the Plk1 or Nek7 phosphorylation-site motif using the *Scansite* algorithm. Note that the lower the score, the better the phosphorylation site sequence matches the optimal motif, with a score of 0 indicating a perfect match. Dotted lines indicate the 25% cut-off for best scoring sites. (**E**) Purified wild-type (WT) or kinase-dead (KD) Plk1 or Nek7 was incubated with HA-tagged Cdc27 WT or S434A/S435A double mutant (SS/AA), followed by phosphorimaging (32P) and immunoblotting for kinase and Cdc27 substrate levels. To quantify the level of phosphorylation ($^{32}$P/HA), the $^{32}$P signals were corrected for HA signal, and the resulting value for the WT kinase with WT substrate sample (lane 3) was set to 1. (**F**) As in panel (**E**), now using RacGAP1 WT or S170A mutant as substrate. (**G**) A model of the signaling interactions and common phosphorylation site motif shared between Plk1, Nek9, Nek6, Nek7 that result in potential phospho-motif amplification during mitosis.

DOI: https://doi.org/10.7554/eLife.44635.018

The following figure supplement is available for figure 7:

*Figure 7 continued*

**Figure supplement 1.** Preparations of isolated Nek kinases are free of Plk1.
DOI: https://doi.org/10.7554/eLife.44635.019

Plk1 substrates (*Kraft et al., 2003*; *Burkard et al., 2009*). To test whether Nek7 could also directly phosphorylate these substrates, we generated WT Cdc27 and S434A/S435A double mutant proteins using *in vitro* transcription/translation (IVTT), and incubated them with isolated Plk1 or Nek7 in an *in vitro* kinase assay. As previously reported, Cdc27 was readily phosphorylated by WT but not KD Plk1, and this phosphorylation was reduced by mutation of the S434/S435 target site (*Figure 7E*, upper panels; *Kraft et al., 2003*). Interestingly, the same assay performed with Nek7 instead of Plk1 demonstrated that Nek7 could also phosphorylate Cdc27 on the same site as Plk1 (*Figure 7E*, lower panels). A second *in vitro* kinase assay using RacGAP1 confirmed that this substrate is also directly phosphorylated by both Plk1 and Nek7, although phosphorylation was not reduced by mutation of S170, indicating that both kinases appear to target sites other than or in addition to S170 (*Figure 7F*). Together, these experiments indicate that Plk1 and Nek6/7/9 share a strikingly similar phosphorylation-site motif, have substantial overlap in potential substrates in the phosphoproteome, and are capable of phosphorylating the same protein substrates on the same target sites *in vitro*.

## Discussion

While it is widely appreciated that the Nek kinases function in various stages of mitosis, and also play roles in essential non-mitotic processes including cilia formation and the DNA damage response (*Moniz et al., 2011*; *Fry et al., 2017*), only a limited number of direct substrates of these kinases have been identified to date. To facilitate future substrate identification, we systematically determined the phosphorylation-site motif for each kinase in the family, with the exception of Nek11, using OPLS. This analysis revealed that all Nek kinases share a common preference for selected hydrophobic residues in the −3 position, but differ substantially in their preference for specific amino acids at other positions within the motif, including their preference for either a threonine (Nek1/3/4/5/8) or serine (Nek2/6/7/9) as the phospho-acceptor residue. A strong determinant of phospho-acceptor site specificity is the residue in the activation loop of the kinase at the DFG+1 position (*Chen et al., 2014*), and the observed dichotomy of serine versus threonine phospho-acceptor preference within the Nek family can be explained by the presence of a leucine versus isoleucine/serine in this position (*Figure 6C*).

Overall, the phosphorylation-site motifs indicate that the Nek family can be divided in four specificity-groups (*Figures 6D* and *7A*). To date, all kinases in specificity Groups 3 (Nek2/10) and 4 (Nek6/7/9) have been implicated in one or more mitotic processes, with the exception of Nek10. The exact function of Nek10 is not known, but it has previously been reported to play a role in the response to UV-stress (*Moniz et al., 2011*). It is therefore tempting to speculate that the Nek kinase specificity clustering might reflect functional divergence between mitotic and non-mitotic Nek kinases.

In addition to the four specificity-groups, we identified amino acid selection preferences that are unique to individual Nek kinases. For example, Nek6 and Nek7 showed a clear preference for substrate peptides that are already phosphorylated on tyrosine on the −1 position, suggesting crosstalk between tyrosine and serine/threonine kinases during mitosis. It is becoming increasingly clear that tyrosine phosphorylation events do, in fact, play a role in mitotic signaling (*Caron et al., 2016*), and multiple tyrosine kinases have been shown to prefer an acidic residue in the −1 position (*Miller and Turk, 2018*), which would generate good target sites for Nek6/7 (*i.e.* a DE-pY-S motif).

In general, good substrates for a kinase usually contain phosphorylation site sequences that closely match the optimal sequence motif for the kinase, although additional factors including secondary docking sites, surface accessibility, and substrate protein conformation also play important roles (*Linding et al., 2007*; *Miller and Turk, 2018*). It is therefore expected that the phosphorylation sites on Nek kinase substrates should be enriched for the amino acids preferred by the upstream Nek kinase. However, to date, too few substrates have been definitively and reproducibly characterized for most of the individual Nek kinases to allow quantitative comparison of those sites with the optimal motifs. The maximal number of substrates reported for any Nek kinase is eleven (for Nek6),

with less than five substrates known for all other Nek kinases except for Nek2. There are no published substrates for Nek 4, 5, 8, and 10. Nonetheless, a global analysis of all previously reported Nek kinase target sites (*Supplementary file 1*) reveals an overrepresentation of leucine and phenylalanine in the −3 position, consistent with the motif reported here. Of the eleven published Nek6 substrates, three contain an asparagine on the −2 position, which we identified as a specificity unique to Group 4 Nek kinases, and four others have a glycine, which is also positively selected for by Nek6. However, not all reported phosphorylation sites on Nek kinase substrates match the optimal Nek kinase motifs, and some even contain strongly disfavored residues such as a proline in the + 1 position (*Supplementary file 1*). This could indicate that such a target site is not directly phosphorylated by a Nek kinase, even though its phosphorylation status was affected by manipulation of Nek kinase activity. However, it is important to emphasize that substrate phosphorylation by a kinase does not require a perfect match between the target site and the optimal motif. Rather, the better the site matches the optimal motif, the more efficiently it is phosphorylated by the kinase (*Obenauer et al., 2003*), as we also demonstrate here in our peptide phosphorylation studies (*Figure 2B-F*).

Importantly, our assays revealed that Nek10 is a dual-specificity kinase, and thus joins a small group of human serine/threonine kinases with reported dual-specificity. These are the GSK-3 kinases, the DYRK-kinases, the kinases of the MAP2K-family, TTK, CK2, Myt1 and the TESK-kinases (*Gómez and Cohen, 1991*; *Lindberg et al., 1993*; *Wang et al., 1994*; *Kentrup et al., 1996*; *Liu et al., 1997*; *Wilson et al., 1997*; *Toshima et al., 1999*). GSK-3 and the DYRK kinases only phosphorylate tyrosine residues when auto-phosphorylating during folding *in cis*, although it was recently reported that the folded, mature DYRK1A retains a weak tyrosine auto-phosphorylation capability (*Cole et al., 2004*; *Lochhead et al., 2005*; *Soundararajan et al., 2013*). TTK and CK2 have weak tyrosine phosphorylation activity compared to their serine phosphorylation activity, and Myt1 and the kinases of the MAP2K family only phosphorylate on tyrosine and threonine in the very distinct context of a TY-site on Cdk1 and a TxY-site on MAP kinases, respectively (*Gómez and Cohen, 1991*; *Lindberg et al., 1993*; *Liu et al., 1997*; *Marin et al., 1999*). The TESK-kinases are therefore an exception within the group of dual-specificity kinases, because they can efficiently phosphorylate multiple protein substrates *in trans* on both tyrosine and serine (*Toshima et al., 1999*).

Here, we show that Nek10's dual-specificity is most like that of the TESK-kinases, because Nek10 phosphorylates tyrosine residues as efficiently as serine residue on substrates *in trans*. Serine/threonine kinases are structurally different than tyrosine kinases, primarily in the catalytic loop and the P+1 loop of the kinase domain. These regions make extensive contact with the substrate, including with the phospho-acceptor site, and structural differences within these loops dictate whether kinases exclusively phosphorylate either serine/threonine or tyrosine (*Taylor et al., 1995*). We show here that an isoleucine in the APE-4 position, and a threonine in the HRD+2 position, makes the P+1 loop and catalytic loop of Nek10 unlike those typically found in either tyrosine kinases or serine/threonine kinases, and these unique residues give Nek10 the capacity to phosphorylate substrates on tyrosine in addition to serine. Serine/threonine kinases have a conserved lysine on the HRD+2 position in the catalytic loop, which makes contact with the P+1 loop and the substrate backbone, and therefore helps positioning the substrate for phosphorylation. The HRD+2 threonine, rather than lysine, in Nek10 is essential for its ability to phosphorylate substrates on tyrosine, and within the dual-specificity kinome this residue is only shared with the TESK kinases. This could potentially explain why of all dual-specificity kinases, Nek10 and the TESK kinases appear to have the broadest tyrosine phosphorylation capacity. Although our experiments with a Nek1 HRD+2 mutant suggest that an HRD+2 threonine is not sufficient for dual-specificity in all sequence contexts, it would be interesting to study the phospho-acceptor site specificity of the serine/threonine kinases BubR1, Plk4, PRPK and MISR2, which also have a serine or threonine residue in the HRD+2 position.

Additionally, the sequence surrounding the phosphorylation site also affects substrate positioning, and can therefore affect phospho-acceptor site specificity. As an example of this interdependence, we have previously noted that the preference of the kinases ATM/ATR for phosphorylating serine-glutamine versus threonine-glutamine motifs appears to depend on the residues found at other positions in the motif (*Joughin et al., 2012*). A similar observation of interdependence between phospho-acceptor residue identity and specific amino acids in other positions within the motif has been shown for the dual-specificity kinase CK2 (*Marin et al., 1999*). Marin et al. observed that tyrosine phosphorylation of a yeast substrate of CK2 depended on specific residues in the −1

and +1 positions. However, phosphorylation of the same substrate with the tyrosine replaced by a serine residue was considerably less influenced by substitutions of the amino acids in the −1 and +1 positions and instead depended on the residue identity in the +3 position (*Marin et al., 1999*). Here, we identified distinct phosphorylation-site motifs for Nek10 on tyrosine and on serine/threonine substrates (*Figure 2*). This indicated that the ability of Nek10 to phosphorylate peptides on tyrosine was strongly dependent on the presence of an aromatic residue or a leucine in the +1 position of the substrate (*Figure 2C*), but Nek10 was much more tolerant of other amino acids in this position when phosphorylating substrates on serine (*Figure 2A*). Potentially, a bulky +1 residue orients the substrate within the catalytic cleft in a manner that generates space for the large tyrosine phospho-acceptor site.

Finally, our study showed that Nek6/7/9 share a common phosphorylation motif with Plk1. Plk1 interacts with Nek9 and functions as the upstream activating kinase in a linear mitotic signaling pathway with Nek6, 7, and 9, raising the possibility that the common phosphorylation motif we observed arises from cross-contamination in the isolated kinase preparations. However, the fact that our Plk1 motif was determined using the recombinant kinase domain of Plk1 expressed in bacteria, and that catalytically dead versions of each of these Nek kinases displayed no phosphorylation activity (*Figure 1—figure supplement 2*) argues against this interpretation. Instead, it may be that these mitotic kinases function together to form a coherent feed-forward loop analogous in some ways to the coherent type 1-feed forward loop with OR gate function described by Uri Alon for transcription factor signaling (*Alon, 2007*). Such a signaling network of kinases sharing a highly related phosphorylation-site motif (*Figure 7G*) might allow persistence of this phospho-motif for a period of time after Plk1 inactivation, or could help spread this specific phospho-motif beyond direct Plk1 substrates by acting as a 'phospho-motif amplifier'. Clearly, many future experiments investigating the spatial and temporal phosphorylation of *in vivo* substrates by kinases within this network will be required to distinguish between these and other possibilities.

# Materials and methods

### Key resources table

| Reagent type (species) or resource | Designation | Source or reference | Identifiers | Additional information |
|---|---|---|---|---|
| Cell line (*Homo sapiens*) | HEK 293T | ATCC | ATCC Cat# CRL-3216, RRID:CVCL_0063 | |
| Cell line (*Homo sapiens*) | U-2 OS | ATCC | ATCC Cat# HTB-96, RRID:CVCL_0042 | |
| Cell line (*Homo sapiens*) | U-2 OS Nek10 KO | This paper | | Nek10 gene was edited by CRISPR technique, selected clone |
| Cell line (*Homo sapiens*) | U-2 OS Nek10 KO + ind Nek10 | This paper | | Dox-inducible Nek10 was introduced in the Nek10 KO clone |
| Antibody | Mouse monoclonal anti-FLAG M2 | Sigma | Sigma-Aldrich Cat# F1804, RRID:AB_262044 | WB 1:1000 |
| Antibody | Rabbit monoclonal anti-FLAG | Sigma | Sigma-Aldrich Cat# F7425, RRID:AB_439687 | WB 1:1000 |
| Antibody | Rat monoclonal anti-HA 3F10 | Roche | Roche Cat# ROAHAHA, RRID:AB_2687407 | WB 1:500 |
| Antibody | Mouse monoclonal anti-B-Actin | Sigma | Sigma-Aldrich Cat# A2228, RRID:AB_476697 | WB 1:10,000 |
| Antibody | Rabbit polyclonal anti-B-Actin | Cell Signaling Technology | Cell Signaling Technology Cat# 4967, RRID:AB_330288 | WB 1:1000 |

*Continued on next page*

*Continued*

| Reagent type (species) or resource | Designation | Source or reference | Identifiers | Additional information |
|---|---|---|---|---|
| Antibody | Rabbit monoclonal anti-phosphotyrosine | Cell Signaling Technology | Cell Signaling Technology Cat# 8954, RRID:AB_2687925 | WB 1:2000 |
| Antibody | Rabbit polyclonal anti-Plk1 | Santa Cruz Biotechnology | Santa Cruz Biotechnology Cat# sc-5585, RRID:AB_2167406 | WB 1:100 |
| Antibody | Mouse monoclonal anti-Plk1 | Santa Cruz Biotechnology | Santa Cruz Biotechnology Cat# sc-17783, RRID:AB_628157 | WB 1:100 |
| Antibody | Goat polyclonal - anti-Mouse 800/680 | LI-COR Biosciences | LI-COR Biosciences Cat# 926–32210, RRID:AB_621842 | WB 1:10,000 |
| Antibody | Goat polyclonal- anti-Rabbit 800/680 | LI-COR Biosciences | LI-COR Biosciences Cat# 926–32211, RRID:AB_621843 | WB 1:10,000 |
| Antibody | Goat polyclonal- anti-Rat 800 | LI-COR Biosciences | LI-COR Biosciences Cat# 926–32219, RRID:AB_1850025 | WB 1:10,000 |
| Antibody | Anti-FLAG M2 affiinity gel | Sigma | Sigma-Aldrich Cat# A2220, RRID:AB_10063035 | |
| Antibody | Anti-HA affinity matrix | Roche | Roche Cat# 11815016001, RRID:AB_390914 | |
| Recombinant DNA reagent | V1900-Nek3-3xFLAG (idem for Nek4-Nek9) | This paper | | Gateway cloned from ORFeome into V1900 (pCMV-Sp6 with C-term. FLAG-tag) |
| Recombinant DNA reagent | pcDNA3-3xFLAG | This paper | | A 3xFLAG primer dimer was cloned by restrion ligation into pcDNA3 for C- or N-terminal tagging |
| Recombinant DNA reagent | pcDNA3-Nek1-3xFLAG | This paper | | PCR NEK1 from ORFeome, cloned into pcDNA3-3xFLAG |
| Recombinant DNA reagent | pcDNA3-Nek4 Cat.-3xFLAG | This paper | | PCR Nek4 Catalytic domain from V1900-Nek4, cloned into pcDNA-3xFLAG |
| Recombinant DNA reagent | pcDNA3-MBP-3xFLAG | This paper | | PCR maltose binding protein, cloned into pcDNA-3xFLAG (N-term.) |
| Recombinant DNA reagent | pcDNA3-MBP-3xFLAG-Nek5 | This paper | | PCR Nek5 Catalytic domain from V1900-Nek5, cloned into pcDNA-MBP-3xFLAG |
| Recombinant DNA reagent | pcDNA3-3xFLAG-TEV-2xstrep-6xHIS | This paper | | A gene block with the complete tag was cloned into pcDNA3 |

*Continued*

| Reagent type (species) or resource | Designation | Source or reference | Identifiers | Additional information |
|---|---|---|---|---|
| Recombinant DNA reagent | pcDNA3-Nek2-3x FLAG-TEV-2xstrep-6xHIS | This paper | | Nek2 Catalytic domain PCR from copy-DNA, cloned into pcDNA3-3x FLAG-TEV-2xstrep-6xHIS |
| Recombinant DNA reagent | pCMV-3xFLAG-Nek10 | PMID: 20956560 | | |
| Recombinant DNA reagent | pcDNA-3xFLAG-Nek10 | this paper | | PCR correction of missing cDNA fragment, cloning into pcDNA3 |
| Recombinant DNA reagent | pET-RSF-Plk1 Cat Domain | This paper | | PCR Cat domain Plk1, cloning into pET-RSF |
| Recombinant DNA reagent | pcDNA-2xHA-Strep | This paper | | pcDNA3 modified with a 2xHA-1xStrepTAGII |
| Recombinant DNA reagent | pcDNA-2xHA-Strep-Cdc27 | This paper | | PCR Cdc27 from copy-DNA cloe into pcDNA-2xHA-Strep |
| Recombinant DNA reagent | pcDNA-2xHA-Strep-RacGAP1 | This paper | | PCR RacGAP1 from copy-DNA cloe into pcDNA-2xHA-Strep |
| Recombinant DNA reagent | LentiGuide-Puro | Addgene 52963 | RRID:Addgene_52963 | |
| Recombinant DNA reagent | LentiGuide-Puro Nek10 sgRNA | This paper | | GTCTGAGCCCG CCATCAGGG |
| Recombinant DNA reagent | LentiGuide-Puro Cas9 sgRNA | This paper | | CTTGTACTCGT CGGTGATCA |
| Recombinant DNA reagent | pLVX-Cas9-Zsgreen | This paper | | Cas9 from addgene 50661 into pLVX-ZsGreen |
| Recombinant DNA reagent | pX335-Cas9(D10A) | Addgene 42335 | RRID: Addgene_42335 | |
| Recombinant DNA reagent | pCW-3xFLAG-Nek10-T2A-eGFP | This paper | | Replace Cas9 in addgene 50661 with Nek10-T2A-eGFP, replace puromycin with blasticidin cassette |
| Peptide, recombinant protein | 3xFLAG peptide | APExBIO | 3xFLAG peptide_ APExBIO:A6001 | |
| Peptide, recombinant protein | Kinase substrate library group I | Anaspec | Kinase substrate library group I_ Anaspec:AS-62017–1 | |
| Peptide, recombinant protein | Kinase substrate library group II | Anaspec | Kinase substrate library group II_Anaspec:AS-62017–1 | |
| Peptide, recombinant protein | Kinase substrate library 0 column S/T/Y | Anaspec | Custom order | |
| Peptide, recombinant protein | Tyrosine peptide library | JPT | Custom order | |
| Chemical compound, drug | Odyssey blocking buffer | LI-COR Biosciences | Odyssey blocking buffer in PBS_ LI-COR:927–40000 | |
| Software, algorithm | Morpheus | https://software.broad institute.org/morpheus/ | | |

*Continued on next page*

*Continued*

| Reagent type (species) or resource | Designation | Source or reference | Identifiers | Additional information |
|---|---|---|---|---|
| Other | Pintool | V and P Scientific | Tube style floating pin_V and P Scientific:FP3S200 | |
| Other | Mantis nanodispenser | Formulatrix | Manti liquid handler_Formulatrix | |
| Other | p81 phospho cellulose paper | Reaction biology | P81 Ion exchange celluose chromatography paper_Reaction Biology Corp | |
| Other | Strep-Tactin XT superflow | IBA lifesciences | Strep-Tactin XT superflow_IBA Life sciences:2–4010 | |
| Other | Nickel-column | GE healthcare | Ni Sepharose 6 Fast flow_GE healthcare: 17531801 | |
| Other | Streptavidin membrane | Promega | SAM2 Biotin capture mebrane_ Promega:V686X | |

## Cloning

Coding DNA sequences of Nek1 and Nek3-Nek9 were obtained from the CCSB-Broad Human Kinase ORF collection. Using gateway cloning technology, Nek3-Nek9 were cloned into V1900 (a kind gift from Karen Colwill/Tony Pawson, The Lunenfeld-Tanenbaum Research Institute, Toronto, ON, Canada) a gateway destination vector derived from pCMV-Sp6 with a C-terminal 3xFLAG epitope tag (DYKDHDGDYKDHDIDYKDDDDK). All other cloning was done by restriction ligation, Quick-Fusion cloning (Biotool.com) or Gibson assembly using the NEBuilder HiFi DNA assembly Cloning Kit (New England Biolabs). Nek1 was PCR amplified from the ORFeome entry vector, and cloned into pcDNA3 that was re-engineered to contain a C-terminal 3xFLAG tag (pcDNA3-MCS-3xFLAG). The Nek4 kinase domain (aa: 1–300) was PCR amplified from the V1900-Nek4 plasmid, and cloned into pcDNA3-MCS-3xFLAG. The Nek5 kinase domain (aa: 2–300) was PCR amplified from the V1900-Nek5 plasmid, and cloned into pcDNA3, which was reengineered with an N-terminal MBP (maltose-binding-protein)–3xFLAG epitope tag. The Nek2 kinase domain (aa: 1–271) was PCR amplified from copy-DNA generated from U-2 OS mRNA, and cloned into pcDNA3, which was reengineered with a C-terminal 3xFLAG-TEV-2xStreptagII-6xHIS epitope tag. (TEV: Tobacco Etch Virus cleavage site (ENLYFQG), StreptagII: WSHPQFEK). The construct pCMV-3xFLAG-Nek10 has been described previously (*Moniz et al., 2011*). A small missing region in the Nek10 cDNA (bp: 2869–3012) was repaired by PCR, and the cDNA including N-terminal 3xFLAG-tag was subcloned into pcDNA3, or the cDNA without 3xFLAG-tag was subcloned into pcDNA3-2xHA-Strep, which is a pcDNA3-variant that was reengineered for epitope-tagging with an N-terminal 2xHA-StreptagII epitope tag (HA: YPYDVPDYA). The Plk1 kinase domain (aa: 38–346) was PCR amplified from plasmids previously described and cloned into pET-RSF (*Elia et al., 2003*). Cdc27 and RacGAP1 cDNA was PCR amplified from copy-DNA, and cloned into pcDNA3-2xHA-Strep. All mutants of the Neks, and Cdc27 and RacGAP1 mutants, were generated by PCR-mediated mutagenesis. All kinase-dead variants were made by mutation of the aspartic acid in the conserved HR<u>D</u> motif into an asparagine, and in case of Nek4, an additional aspartic acid to asparagine mutation was made in the conserved <u>D</u>FG-motif (DLG in case of Nek4). In order to make the Nek10 KO cell-line, a Nek10-targeting sgRNA (sequence: GTCTGAGCCCGCCATCAGGG) was cloned into lentiGuide-Puro (addgene #52963), and Cas9 was subcloned from pCW-Cas9 (addgene #50661) into pLVX-Cas9-ZsGreen. In addition, a Cas9-targeting sgRNA (sequence: CTTGTACTCGTCGGTGATCA) was cloned into lenti-guide-puro. The inducible Nek10 construct was generated by replacing Cas9 in pCW-Cas9 with

3xFLAG-Nek10-T2A-eGFP. Silent mutations were introduced in the PAM-sequence of the Nek10 sgRNA target site, and the puromycin cassette in pCW was replaced with a blastidicin resistance cassette.

## Protein production and purification

All Nek kinases, unless indicated otherwise below, were purified from transiently transfected HEK 293T cells. In short, $7 \times 10^6$ HEK 293T cells were plated in 15 cm plates, and the next day trans-fected with 20–25 µg kinase-DNA using polyethyleneimine (PEI). After O/N incubation, transfection medium was replaced with fresh medium, and another 24 hr later cells were washed with PBS and harvested by scraping in lysis buffer (20 mM Tris-HCl pH 7.5, 150 mM NaCl, 1 mM EDTA, 1 mM EGTA, 1% Triton-X100, 1 mM DTT) supplemented with protease inhibitors (complete EDTA-free protease inhibitor cocktail, Roche) and phosphatase inhibitors (PhosSTOP, Roche). Lysates were incubated on ice for 20 min, and cleared by centrifugation. Next, anti-FLAG M2-affinity gel (Sigma) was added to the cleared lysate, followed by incubation for 2 hr at 4˚C while rotating. After incuba-tion, the beads were pelleted by centrifugation, washed twice with lysis buffer and washed twice again with wash buffer (50 mM HEPES pH 7.4, 100 mM NaCl, 1 mM DTT, 0.01% NP-40, 10% glyc-erol). Protein was eluted from the beads by incubation for 1 hr at RT (while rotating) with 150–250 µl wash buffer supplemented with phosphatase inhibitors (PhosSTOP, Roche) and 0.5 mg/ml 3xFLAG peptide (APExBIO). To determine concentration and purity, a BSA standard curve and a sample of the eluate were analyzed by SDS-PAGE and Coomassie-staining (SimplyBlue Safe stain, Life Technol-ogies). Eluted kinase was either used directly, or snap-frozen and stored at −80˚C. The 2xHA-Strep-tag-Nek10 KD (HA-tagged Nek10 KD, Figure 5) was isolated essentially exactly as described above. However, Strep-Tactin XT superflow (IBA lifesciences) was used instead of anti-FLAG affinity gel, the composition of the wash buffer was different (100 mM Tris pH 8, 150 mM NaCL, 1 mM EDTA) and the kinase was eluted in 1x Biotin elution buffer BXT (IBA lifesciences). Plk1 was isolated from bacte-ria. A BL21 Rosetta two strain was transformed and induced to express the 6xHIS-MBP-TEV-tagged Plk1 kinase domain. The bacteria were lysed, followed by incubation with an Amylose resin (New England Biolabs)) and elution with maltose. In case of constitutively active T210D Plk1, a second purification step on a nickel-column (Ni Sepharose 6 Fast Flow, GE Healthcare) was performed, fol-lowed by imidazole-elution.

## Oriented Peptide Library Screen

Oriented Peptide Library Screens were performed as described in Hutti et al., but with a slightly adapted protocol so that the assay could be done in 1536-well plates (Hutti et al., 2004). The ser-ine/threonine peptide library (Kinase Substrates Library, Group I and Group II, Anaspec) consists of 198 peptide pools with the sequence Y-A-Z-X-X-X-X-S/T-X-X-X-X-A-G-K-K-(LC-Biotin)-NH2, were X = mixture of all natural amino acids except Cys, Ser and Thr, S/T = equal mixture of Ser and Thr, and LC-Biotin = long chain biotin. In each peptide pool, the Z is fixed to be a single amino acid, which can be any of the 20 natural amino acids or phospho-Thr or phospho-Tyr. In the example, the Z is positioned on the P-5 relative to the S/T, but in the complete library the Z would be fixed in a position ranging from −5 to +4 relative to the S/T. Hence, there are nine different positions and 22 different amino acids, generating 198 different peptide pools. In addition, the library contains three peptide pools that are completely degenerate with the exception of a fixed Ser, Thr or Tyr as phos-pho-acceptor site for use in determining the 0 position (Figure 1—figure supplement 1A). The tyro-sine-peptide library (custom order, JPT) consisted of peptide pools with the sequence G-A-Z-X-X-X-X-Y-X-X-X-X-A-G-K-K-(LC-Biotin)-NH2, and in this library Cys, Ser, Thr and Tyr were absent from the amino acid mixtures on the X positions. All peptides were dissolved in DMSO to 7.5 mM for the Ser/Thr library, and 5 mM for the Tyr-library. The peptides were arrayed in a 1536-well plate by multi-channel pipetting, and diluted to 500 µM with water.

For the kinase assay, 2.2 µl kinase buffer per well was dispensed in 1536-well plates using a Man-tis nanodispenser robot (Formulatrix). The core kinase buffer consisted of 50 mM Tris-HCl pH 7.5, 0.1 mM EGTA, 1 mM DTT, 50 µM ATP, 1 mM PKA-inhibitor (Sigma) and 0.1% Tween-20, and was supplemented with 10 mM MgCl2 and 2 mM MnCl2 for Nek1/2/3/4/6/7/9, 10 mM MnCl2 for Nek5/8, and 10 mM MgCl1 and 10 mM MnCl2 for Nek10. Next, 250 nl peptide was transferred from the peptide-stock plate to the kinase assay plate using a pintool with slot tips (V and P Scientific, Pin

type FP3S200). Subsequently, 200 nl of a mix of kinase and [γ-$^{33}$P]-ATP was added using the Mantis, with a final concentration in each well ranging from 5 to 50 nM of the kinase and 0.027 μCi/μl for the [γ-$^{33}$P]-ATP. Between every step, the 1536-well plate was centrifuged briefly to collect all liquid, and cooled on ice as needed to prevent evaporation. After addition of the kinase, the plate was incubated at 30℃ for 2 hr. Subsequently, 250 nl of each reaction was spotted on streptavidin-membrane (Promega) using the pintool, the membrane was washed and dried, exposed to a phosphorimaging screen, and imaged on a Typhoon 9400 imager (GE Healthcare).

Spot intensities were quantified using ImageStudio (Li-COR BioSciences), and normalized by dividing the measured intensity for an individual spot by the average spot intensity for all amino acids in that position. For each kinase, OPLS-experiments were done at least twice, and normalized values were averaged for all replicates. In case of a fixed serine or threonine on the Z-position, the fixed residue presents a second acceptor site in addition to the 0-position S/T, and spot intensities are therefore high at those positions, which does not necessarily reflect real sequence preferences. To correct for this, normalized intensities for fixed S or T were changed to one in case of intensity >1. For Nek10 on the Tyr-library, the same was done but for intensities measured for fixed-Tyr peptides. The exception was the +1 Tyr, because we reasoned the normalized intensity >1 most likely reflected a true preference, considering its strong preference for aromatic amino acids. Next, normalized values were log2-transformed. These values were used to generate heatmaps using Morpheus (https://software.broadinstitute.org/morpheus/). These values were transformed to sequence logos with Seq2Logo (*Thomsen and Nielsen, 2012*).

### *In vitro* kinase assays

For *in vitro* kinase assays with a peptide-substrate, 5–50 nM of kinase was pre-incubated at 30℃ for 30 min in the same kinase buffer as used for OPLS, but without the 0.1% Tween-20. Subsequently, [γ-$^{33}$P]-ATP (or in some cases [γ-$^{32}$P]-ATP) was added to a final concentration of 0.075–1 μCi/μl, and peptide added to a final concentration of 50 μM. Peptides were synthesized in-house at the 5 μmol starting resin scale per 96-well plate well, using an MultiPep small scale peptide synthesizer (Intavis) and standard FMOC chemistry. In addition to the peptide-specific 10-mer sequences indicated in the figure panels, each peptide had an N-terminal Y-A (or W-A for peptides of Nek10) sequence to allow concentration determination of peptide-solutions by UV-spectrophotometry. Also, each peptide had a C-terminal G-K-K-K sequence to increase solubility and interaction with phosphocellulose. After adding substrate and [γ-$^{33}$P]-ATP, the mixture was incubated at 30℃ for 20 min, and every 5 min a sample was spotted on p81 phosphocellulose papers (Reaction Biology Corp.), with the exception of some of the repeats with the Nek10 T693P variant, that were incubated for 1 hr or 2 hr, with spotting every 20 min or 30 min respectively. After spotting, the phosphocellulose papers were immediately soaked in 0.5% phosphoric acid. At the end of the assay, all papers were washed three times in 0.5% phosphoric acid, air-dried, transferred to vials with scintillation counter fluid and counted by a LS 6500 scintillation counter (Beckman Coulter). The resulting data points were analyzed by linear regression, and the phosphorylation rate was defined as the slope of the linear regression curve.

To study Nek10 auto-phosphorylation, 250 ng kinase was incubated in OPLS-kinase buffer with 1 μCi/μl [γ-$^{32}$P]-ATP for 30 min at 30℃. The reaction was stopped by adding 6x sample buffer (SB) for SDS-PAGE (208 mM Tris-HCl pH 6.8, 42% glycerol, 3 mM beta-mercapto-ethanol, 10% SDS, 5 mg/ml bromophenol blue) and boiling for 5 min at 95℃. The sample was split, and analyzed by SDS-PAGE followed by Coomassie staining to asses total Nek10 levels, or analyzed by SDS-PAGE followed by phosphorimaging to analyze protein phosphorylation.

To study Nek10 auto-phosphorylation *in trans*, 250 ng of FLAG-tagged kinase-proficient Nek10 was mixed with 250 ng of HA-tagged kinase-deficient Nek10 and 0.06 μCi/μl [γ-$^{32}$P]-ATP in kinase buffer. The mixture was incubated at 30℃ for 15 min to 2 hr, depending on the assay, and the reaction was stopped by adding lysis buffer supplemented with EDTA to a final concentration of 20 mM. The sample was split in half and incubated for 1 hr at RT (while rotating) with either anti-FLAG M2 affinity gel (Sigma) or anti-HA affinity matrix (mAb 3F10; Roche). The beads were washed once with lysis buffer, dried, and resuspended in 1xSB. Nek10 was eluted of the beads by boiling in 1xSB, the sample was split, and 1/4th was analyzed by SDS-PAGE followed by immunoblotting to asses total Nek10 levels, while 3/4ths was analyzed by SDS-PAGE followed by phosphorimaging to assess protein phosphorylation.

## U-2 OS KO cell-line generation

All cell-lines were grown at 37°C in a humidified incubator supplied with 5% $CO_2$, in DMEM supplemented with 10% FBS. All cell-lines were tested and shown to be free of mycoplasma contamination. Lentivirus was produced by transient transfection of HEK 293T cells using the CalPhos mammalian transfection kit (Clontech laboratories) to introduce packaging vectors (VSVg and GAG/POL/Δ8.2) and either pLVX-Cas9-Zsgreen, or lentiguide-PURO-Nek10 sgRNA, or pCW-Nek10-T2A-GFP-Blasticidin. The viral supernatant was filtered through a 0.45 μm filter and used to transduce U-2 OS target cells. First, U-2 OS cells were transduced with Cas9, followed by sorting for Zsgreen positive cells. These cells were then transduced with Nek10 sgRNA, followed by selection with puromycin (2 μg/ml; Invivogen). Doubly-transduced cells were subsequently cultured for 2 weeks to allow editing, and single cell sorted into 96-well plates. After expansion of the single-cell clones, gene editing was analyzed by PCR-amplifying the edited locus from gDNA, followed by Sanger sequencing of the PCR product, together with TIDE analysis (*Brinkman et al., 2014*).

After selecting a Nek10-KO clone, Cas9 was removed by transfecting with a Cas9-sgRNA construct using lipofectamine 2000 (Thermo Fisher Scientific). Cells were cultured for 10 days and sorted for ZsGreen-negative cells. Immunoblot analysis indicated that although all cells were negative for Zsgreen, about 30% of Cas9-expression still remained. Next, the cells were transduced with the pCW-Nek10 constructs, followed by selection with Blasticidin S (10 μg/ml; Invitrogen). To select for cells without leaky expression of Nek10, but good induction upon treatment with doxycycline, the following was done: first, GFP-negative cells were sorted, followed by induction of Nek10 expression with 2 μg/ml doxycycline (Sigma) for several days, and a second round of sorting, this time on GFP-positive cells.

## Antibodies and western blotting

Cell lysates was analyzed by SDS-PAGE, and transferred to nitrocellulose membranes by tank electrotransfer. Membranes were blocked using either 5% skim milk in Tris-Buffered-Saline (TBS), or with Odyssey blocking buffer (LI-COR BioSciences) for 1 hr at RT. Primary antibody probing was done either in 1% skim milk in TBS-T (TBS with 0.1% Tween-20), or in Odyssey blocking buffer, for 1-2 hr at RT, or O/N at 4°C. Primary antibodies used are Mouse-anti-FLAG M2 (1:1000; Sigma), Rabbit-anti-FLAG (1:1000; Sigma), Rat-anti-HA 3F10 (1:500; Roche), Mouse-anti-β-actin (1:10,000; Sigma), Rabbit-anti-β-Actin (1:1000, Cell Signaling Technology), Rabbit-anti-phospho-tyrosine P-Y-1000 (1:2000; Cell Signaling Technology), Mouse-anti-Plk1 F-8 (1:100; Santa Cruz Biotechnology, Inc) and Rabbit-anti-Plk1 (1:100; Santa Cruz Biotechnology, Inc). After primary antibody incubation, the membranes were washed 3x with TBS-T, followed by staining with secondary antibody in Odyssey blocking buffer for 1 hr at RT. Secondary antibodies used were Goat-anti-Mouse, Goat-anti-Rabbit, or Goat-anti-Rat, all from LI-COR, all 1:10,000, and all labeled with either IRDye 680 or IRDye 800. After secondary antibody staining, the membranes were washed again 3x with TBS-T, and imaged with an Odyssey CLx scanner (LI-COR BioSciences). Image analysis was done using ImageStudio (LI-COR BioSciences).

## Nek10 auto-phosphorylation in cells

To study phosphorylation in cells, U-2 OS cells were transfected with the 3xFLAG-Nek10 vector using lipofectamine 2000 (Thermo Fisher Scientific) for transient expression, or 3xFLAG-Nek10 expression was induced in the stable U-2 OS Nek10 KO cell lines with inducible Nek10 by treatment with docxycycline for 48 hr. The cells were lysed and Nek10 was immobilized on anti-FLAG beads as described in the 'protein purification' section above. Next, beads were washed three times with lysis buffer, and all bound proteins eluted by boiling in 1xSB. To study tyrosine phosphorylation, eluates and total cell lysates were analyzed by SDS-PAGE followed by immunoblotting. To analyze serine phosphorylation by mass spectrometry, eluates were separated by SDS-PAGE, followed by Coomassie staining of the gel. The band containing 3xFLAG-Nek10 was excised, reduced and alkylated, followed by in-gel trypsin digestion. Tryptic peptides were extracted, desalted and lyophilized. Next, phosphopeptides were enriched using the High-Select Fe-NTA phosphopeptide enrichment kit (ThermoFisher Scientific) per manufacturer's instructions. The elution tubes were coated with trypsinized BSA to prevent loss of phospho-peptides from binding to the tube wall. Eluted phosphopeptides were re-suspended in 0.1% acetic acid and separated by reverse phase HPLC using an EASY-

nLC1000 (Thermo) with a self-pack 5 μm tip analytical column (12 cm of 5 μm C18, New Objective). Next, peptides were ionized by nanoelectrospray and analyzed using a QExactive HF-X mass spectrometer (ThermoFisher Scientific). The full MS-scan was followed by MS/MS for the top 15 precursor ions in each cycle. Raw mass spectral data files (.raw) were searched using Proteome Discoverer 2.2 (Thermo) and Mascot version 2.4.1 (Matrix Science). In addition, for the phosphopeptide corresponding to N346-R365 of Nek10, the spectrum was manually annotated by comparison to predicted products using Protein Prospector (http://prospector.ucsf.edu/prospector/mshome.htm) to ensure correct phosphorylation site identification.

For the Mass Spectrometry experiment depicted in *Figure 3—figure supplement 1* , Nek10-deficient HEK 293T cells were generated using CRISPR/Cas9-mediated mutagenesis. In short, intronic sequences surrounding exon 24 of Nek10 (containing the DFG motif) were targeted by transfection with the Cas9(D10A) nickase and sgRNAs with the sequences TAAGGTATCATGCCCTCAT and CTACACTTCATAACCAA in the pX335 plasmid (Addgene 42335). Next, transfected cells were selected with puromycin, followed by expansion of single-cell clones, and PCR screening for deletion of exon 24. These cells were transfected to express FLAG tagged Nek10, followed by anti-FLAG IP and elution with anti-FLAG peptide. Nek10 was incubated in kinase buffer with or without ATP, and analyzed by Mass Spectrometry.

## Motif analysis and substrate prediction

The hierarchical clustering based on the OPLS-datasets was done using the averaged, normalized intensities, with the values for fixed Ser and Thr (except in the 0 position) set to one if >1 (also see OPLS section). Hierarchical clustering was done using Morpheus (https://software.broadinstitute.org/morpheus/) according to a One minus Pearson correlation metric with an average linkage method. For the clustering based on sequence, Nek kinase domain amino acid sequences were uploaded into JalVIEW, aligned with ClustalWS, and the % sequence identity used to derive a dendrogram and similarity matrix.

To predict substrates of Plk1 and Nek6/7/9, we took the averaged, column-normalized intensities of the OPLS-experiments to generate a Position-Specific Scoring Matrix (PSSM), and scored every phosphorylated site in the *Phosphositeplus* database for their match to the PSSM using *Scansite* (*Obenauer et al., 2003*; *Hornbeck et al., 2015*). A Venn diagram was generated for the 10% best (i.e. lowest) scoring sites using the Venn diagram maker online (https://www.meta-chart.com/venn#/display). To predict mitotic substrates of Plk1 and Nek7, Table S2 from *Santamaria et al. (2011)* was used. The phosphosites that are not Class I phosphosites were removed, and duplicate sites were taken out, generating a hitlist of 694 sites consistent with what was reported in Santamaria *et al.* As above, this list of sites was scored using *Scansite* version four for their match to the Plk1 and Nek6/7/9 PSSM.

## Statistical analysis

All statistical analyses were done using Graphpad Prism. If the experiment contained two datasets, significance was calculated using a Students t-test, either paired or unpaired, and either regular or ratiometric, depending on what was most appropriate given the experimental conditions. If the experiment contained more than two datasets, an ANOVA with *post hoc* multiple comparison was performed.

## Acknowledgements

We thank Dan Lim and other members of the Yaffe laboratory for helpful discussion and advice, Forest White (Massachusetts Institute of Technology, Cambridge, MA, USA) for advice on Mass Spectrometry experiments and Prasad Jallepalli (Memorial Sloan Kettering, New York, NY, USA) and Karen Colwill/Tony Pawson (The Lunenfeld-Tanenbaum Research Institute, Toronto, ON, Canada) for kindly sharing reagents. We also thank the Biopolymers and Proteomics core facility, and Genomics: High throughput Screening core facility of the Swanson Biotechnology Center for their assistance. This research was supported by National Institute of Health grants R01-GM104047 (MBY and BET), R01-ES015339 (MBY), R35-ES028374 (MBY), by the Charles and Marjorie Holloway Foundation (MBY), by the MIT Center for Precision Cancer Medicine, by the European Research Council under the European Union's Seventh Framework Program, grant number FP/2007–2013 (ERC grant

KINOMEDRIFT to RL), by the National Cancer Institute of the National Institutes of Health under Award Number K99CA226396 (PC) and by fellowships for BvdK from the Dutch Cancer Society (BUIT 2015–7546) and Ludwig Center Fund. Support was also provided by the Cancer Center Support Grant P30-CA14051 from the National Cancer Institute and the Center for Environmental Health Sciences Support Grant P30-ES002109 from the National Institute of Environmental Health Sciences.

## Additional information

### Funding

| Funder | Grant reference number | Author |
|---|---|---|
| Ludwig Institute for Cancer Research | | Bert van de Kooij |
| Dutch Cancer Society | BUIT 2015-7546 | Bert van de Kooij |
| National Cancer Institute | K99CA226396 | Pau Creixell |
| European Union Seventh Framework Programme | FP/2007-2013 ERC grant (KINOMEDRIFT) | Rune Linding |
| National Institutes of Health | R01-GM104047 | Benjamin E Turk<br>Michael B Yaffe |
| The Charles and Marjorie Holloway Foundation | | Michael B Yaffe |
| National Institutes of Health | R01-ES015339 | Michael B Yaffe |
| National Institutes of Health | R35-ES028374 | Michael B Yaffe |
| National Cancer Institute | P30-CA14051 | Michael B Yaffe |
| National Institute of Environmental Health Sciences | P30-ES002109 | Michael B Yaffe |
| MIT Center for Precision Cancer Medicine | | Michael B Yaffe |

The funders had no role in study design, data collection and interpretation, or the decision to submit the work for publication.

### Author contributions

Bert van de Kooij, Conceptualization, Resources, Data curation, Formal analysis, Funding acquisition, Validation, Investigation, Visualization, Methodology, Writing—original draft, Project administration, Writing—review and editing; Pau Creixell, Data curation, Formal analysis, Investigation, Visualization, Writing—review and editing; Anne van Vlimmeren, Data curation, Formal analysis, Investigation, Writing—review and editing; Brian A Joughin, Data curation, Formal analysis, Writing—review and editing; Chad J Miller, Methodology, Writing—review and editing; Nasir Haider, Resources, Investigation, Writing—review and editing; Craig D Simpson, Investigation; Rune Linding, Conceptualization, Resources, Funding acquisition; Vuk Stambolic, Conceptualization, Supervision, Writing—review and editing; Benjamin E Turk, Conceptualization, Supervision, Funding acquisition, Methodology, Writing—review and editing; Michael B Yaffe, Conceptualization, Resources, Data curation, Formal analysis, Supervision, Funding acquisition, Validation, Investigation, Visualization, Methodology, Writing—original draft, Writing—review and editing

### Author ORCIDs

Bert van de Kooij https://orcid.org/0000-0003-1042-8409
Brian A Joughin https://orcid.org/0000-0003-1022-9450
Vuk Stambolic http://orcid.org/0000-0001-8853-3239
Michael B Yaffe https://orcid.org/0000-0002-9547-3251

Decision letter and Author response
Decision letter https://doi.org/10.7554/eLife.44635.024
Author response https://doi.org/10.7554/eLife.44635.025

## Additional files

### Supplementary files

• Source data 1. PSSMs of Nek family phosphorylation site motifs.
DOI: https://doi.org/10.7554/eLife.44635.020

• Supplementary file 1. List of sites phosphorylated by the Nek kinases. The Phosphositeplus database was mined for sites on human substrates that have been reported to be phosphorylated by any of the Nek kinases for which we determined the motif in this manuscript. The −3 column is shaded yellow. Leucine, methionine and phenylalanine are colored in purple in the −3 and + 1 positions for all Nek kinases, asparigine is colored in red in the −2 position for Nek6, and basic residues are colored in blue in the −1 and +2 positions for Nek1 and Nek3.
DOI: https://doi.org/10.7554/eLife.44635.021

• Transparent reporting form
DOI: https://doi.org/10.7554/eLife.44635.022

### Data availability

The Position Specific Scoring Matrices (PSSMs) containing the quantitative phosphorylation site motif information for each human Nek kinase have been deposited to the database of Scansite 4.0 (http://scansite.mit.edu). They are publicly available to use in all of the features of Scansite, including but not limited to motif prediction on given substrates, or motif-based database searches, simply by selecting the Nek kinase of interest from the dropdown menu presented upon selection of a specific feature. Please see the Scansite tutorial (https://scansite4.mit.edu/4.0/#tutorial) for details. The PSSMs will also be incorporated into the NetPhorest algorithm (https://netphorest.science) and KinomeXplorer platform (https://kinomexplorer.science). The raw PSSMs are available in Source data 1. The data published in Supplementary file 1 was obtained from Phosphositeplus (www.phosphosite.org), and can be accessed directly by performing a substrate search from the Phosphositeplus home page for the Nek kinase of interest. The data presented in Figure 7C was obtained by downloading the complete phosphorylation_site_dataset from Phosphositeplus (https://www.phosphosite.org/staticDownloads), which was analyzed by a custom-built script to score each site for their match to each Nek kinase motif according to the Scansite scoring algorithm.

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
