## [Decision Letter]

Thank you for submitting your article "Comprehensive substrate specificity profiling of the human Nek kinome reveals unexpected signaling outputs" for consideration by *eLife*. Your article has been reviewed by three peer reviewers, and the evaluation has been overseen by a Reviewing Editor, Margaret Frame, and Jonathan Cooper as the Senior Editor.

The reviewers have discussed the reviews with one another and the Reviewing Editor has drafted this decision to help you prepare a revised submission.

Summary:

Mike Yaffe and colleagues describe here the use of oriented peptide library screening to undertake a systematic analysis of the phosphosite motif preference for the human NEK kinome. As well as differentiating subgroups of Neks based on preference for serine versus threonine and specific requirements for charged residues in different positions, they show how Nek10 acts in a unique manner within this family to phosphorylate tyrosine as well as serine. They also present an interesting model to suggest that because the Nek6, Nek7 and Nek9 kinases that work downstream of Plk1 share a very similar motif preference to Plk1, these Neks could possibly act as direct signal amplifiers of Plk1 through phosphorylation of the same substrates.

This is a well-presented manuscript where the data strongly support the conclusions. This work adds valuable understanding to the mechanism of action of kinase-substrate selection by the human NEK kinome, and it will be of interest to the broader kinase community.

The reviewers have suggested the manuscript could be improved by provision of a bit more experimental data, and clarification of a number of points at outlined below.

Essential revisions:

1) Figures 1-4 present experimental data based on phosphorylation of peptide arrays using purified Nek kinases. A Coomassie Blue, or equivalent, stained SDS-PAGE gel should be shown of the purified full-length or catalytic domain fragments of the NEK kinases used for the OPLS screening as part of Figure 1—figure supplement 1. The same samples should also be Western blotted for Plk1 to confirm there no contamination of these preparations with this kinase. This is important in relation to the data presented in Figure 7. In addition, while it is good to see control experiments performed with inactivated versions of each kinase, the authors should note in the text the slight caveat that many NEKs act in cascades with other kinases and that use of a kinase-inactive protein may lead to failure to activate a downstream co-purifying kinase that is responsible for the peptide phosphorylation.

2) The results using mutant kinases to analyze what sequences within its catalytic site give Nek10 the ability to phosphorylate tyrosine are interesting and identify a few very specific residues that appear to make the difference (e.g. APE-4 and HRD+2). It would therefore be worthwhile to make the complementary mutations at these sites in one of the classic serine/threonine Nek kinase, for example Nek1 or Nek7 for which structures have been solved, to see if their activity could be re-directed towards tyrosine.

3) The site of Nek10 phosphorylation. The authors state in subsection “Nek10 is a dual-specificity kinase” and “Nek10 auto-phosphorylates on serine and tyrosine residues in trans, both in vitro and in cells” (latter in relation to Figure 5A and B) that the in vitro sites were identified by mass spectrometry, but the data are 'not shown'. What is the rationale for not revealing these sites in this paper? These should be identified – were they both tyrosine and serine sites? And did they lie within or out-with the catalytic site?

4) Figure 5 reports data using a U-2 OS cell line in which the endogenous Nek10 gene has been removed by gene-editing at least in some experiments. This loss of Nek10 need to be shown with a Western blot or, if good antibodies are not available, by RT-PCR analysis. The 'in cell' data on tyrosine autophosphorylation of Nek10 are unclear; it is not stated whether the U-2 OS cells were WT or Nek10^-/-^ in places (e.g. subsection “Nek10 auto-phosphorylates on serine and tyrosine residues in trans, both in vitro and in cells”), so the contribution of endogenous Nek10 is not always clear.

5) The identification of Y644 as the major site of tyrosine phosphorylation of Nek10 (Figure 5E) is compromised by the knowledge that this mutation at this site decreases kinase activity (Figure 5G) – could the decrease in phosphorylation of the Y644 mutant simply result from decreased protein kinase activity rather than Y644 being the major site of phosphorylation? A similar criticism could be levelled at the observed effect of the Y644 mutation in the trans phosphorylation study (Figure 5F) if the effect of the Y644 mutation was to cause decrease phosphorylation associated with a conformational change that alters the activity of Nek10. Also, the conclusion that Nek10 autophosphorylation can occur in trans in vitro is demonstrated (Figure 5F) and it is reasonable to assume that this phosphorylation in trans may occur in cells, but there is no data to support this conclusion (subsection “Nek10 auto-phosphorylates on serine and tyrosine residues in trans, both in vitro and in cells”). The data and the conclusion supporting this part of the study should be discussed with these points taken into account.

6) Figure 7E and F report that both Plk1 and Nek7 can phosphorylate the Cdc27 and RacGAP1 proteins. It is also stated that mutation of residues that were putative phosphorylation sites for both Plk1 and Nek7 led to reduced phosphorylation. This reduction in phosphorylation should be quantified. In addition, it would help the conclusions if the wild-type substrates (or phosphorylated fragments) were subject to mass spectrometry analysis after incubation with the two kinases to demonstrate that there is significant overlap in the phospho-acceptor sites.

---

## [Author Response]

Essential revisions:1) Figures 1-4 present experimental data based on phosphorylation of peptide arrays using purified Nek kinases. A Coomassie Blue, or equivalent, stained SDS-PAGE gel should be shown of the purified full-length or catalytic domain fragments of the NEK kinases used for the OPLS screening as part of Figure 1—figure supplement 1. The same samples should also be Western blotted for Plk1 to confirm there no contamination of these preparations with this kinase. This is important in relation to the data presented in Figure 7. In addition, while it is good to see control experiments performed with inactivated versions of each kinase, the authors should note in the text the slight caveat that many NEKs act in cascades with other kinases and that use of a kinase-inactive protein may lead to failure to activate a downstream co-purifying kinase that is responsible for the peptide phosphorylation.

As requested, we have analyzed the preparations of the isolated Nek kinases that were used in the OPLS experiments by SDS-PAGE followed by Coomassie staining and added these data as Figure 1—figure supplement 1B. In addition, we have analyzed all of the purified Nek kinase preparations for potential Plk1 contamination by anti-Plk1 western blotting, as suggested. As now shown in the new Figure 7—figure supplement 1, no Plk1 contamination was detected in any of the isolated Nek kinase preparations, although Plk1 could easily be detected in the total HEK 293T cell lysates. We therefore conclude that the Nek6/7/9 motifs that we deduced were not the result of Plk1 contamination of these Nek kinase preparations, which is further supported by the complete absence of peptide library phosphorylation using the same purification techniques for kinase-dead mutants of Nek 6, 7, and 9 (Figure 1—figure supplement 2). Nonetheless, the reviewers were quite correct to point out that there is a minor theoretical possibility that in general, the activity of co-purified kinases might depend on the activity of the ectopically expressed Nek kinase. These new data are now explicitly discussed in subsection “The Nek kinase family diverges into four specificity-groups” of the Results section in the revised manuscript.

2) The results using mutant kinases to analyze what sequences within its catalytic site give Nek10 the ability to phosphorylate tyrosine are interesting and identify a few very specific residues that appear to make the difference (e.g. APE-4 and HRD+2). It would therefore be worthwhile to make the complementary mutations at these sites in one of the classic serine/threonine Nek kinase, for example Nek1 or Nek7 for which structures have been solved, to see if their activity could be re-directed towards tyrosine.

We thank the reviewers for this excellent suggestion. As requested, we have now performed the suggested experiments using Nek1, and included this new data in Figure 3—figure supplement 3. As described in subsection “An APE-4 isoleucine in Nek1 enhances substrate phosphorylation on tyrosine residues” in the revised manuscript, replacement of an isoleucine in the APE-4 position, but not a threonine in the HRD+2 position, increases Nek1 phosphorylation of Tyr-containing peptides (Figure 3—figure supplement 3D,E), and reduces phosphorylation of Thr-containing peptides (Figure 3—figure supplement 3C), albeit while substantially reducing the overall kinase activity (Figure 3—figure supplement 3B). Nonetheless, even in the APE-4 isoleucine mutant, tyrosine phosphorylation is only about 20% as efficient as threonine phosphorylation, indicating that additional sequence determinants in Nek10 collaborate with the APE-4 isoleucine to enhance the dual-specificity nature of the Nek10 kinase domain. Please see the revised manuscript for experimental details and conclusions.

3) The site of Nek10 phosphorylation. The authors state in subsection “Nek10 is a dual-specificity kinase” and “Nek10 auto-phosphorylates on serine and tyrosine residues in trans, both in vitro and in cells” (latter in relation to Figure 5A and B) that the in vitro sites were identified by mass spec, but the data are 'not shown'. What is the rationale for not revealing these sites in this paper? These should be identified – were they both tyrosine and serine sites? And did they lie within or out-with the catalytic site?

As explained in the original manuscript, we mentioned this experiment merely to clarify the basis for the hypothesis that Nek10 might phosphorylate tyrosine residues. The mass spectrometry results were valuable for hypothesis generation, but the experiment was not designed or performed in a comprehensive manner to map all of the sites, and we considered the data too preliminary for publication. (For example, the tryptic fragment containing Y644 is only 4 amino acids long, and was not detected in either the phospho- or non-phospho form). However, based on the reviewers’ request, we now include the results of this mass spectrometry experiment in Figure 3—figure supplement 1 of the revised manuscript, and describe the findings in subsection “Nek10 is a dual-specificity kinase” of the revised text. The reviewers can appreciate that most of the detected phospho-sites in this preliminary experiment lie outside of the kinase domain, with the exception of T670. More importantly however, several of the identified phospho-sites are on tyrosine residues.

4) Figure 5 reports data using a U-2 OS cell line in which the endogenous Nek10 gene has been removed by gene-editing at least in some experiments. This loss of Nek10 need to be shown with a Western blot or, if good antibodies are not available, by RT-PCR analysis. The 'in cell' data on tyrosine autophosphorylation of Nek10 are unclear; it is not stated whether the U-2 OS cells were WT or Nek10^-/-^ in places (e.g. subsection “Nek10 auto-phosphorylates on serine and tyrosine residues in trans, both in vitro and in cells”), so the contribution of endogenous Nek10 is not always clear.

This is a very valid point, and we agree with the reviewers that validating KO cell lines by Western blotting to show loss of protein expression is the preferred method. Therefore, we tested multiple anti-Nek10 antibodies that were commercially available, but unfortunately none of them performed satisfactorily. For this reason we decided to analyze editing of the genomic DNA loci by sequencing of the targeted locus, the results of which were shown in Supplementary Figure 5A of the original and are now shown in Figure 5—figure supplement 1A of the revised manuscript. Note that we sequenced 10 copies of the PCR-amplified Nek10 target site (from 10 bacterial colonies containing a TOPO backbone with the Nek10 PCR insert), and all of them were identified as edited, either by a 19 bp deletion or by a larger deletion that includes the last 80 bp of exon 5. Hence, transcription from both alleles would result in out-of-frame transcripts.

To further address the reviewers’ point, and exclude the small possibility that we missed a WT locus, we have now analyzed the total pool of PCR product of the Nek10 target site by TIDE analysis (Brinkman et al., 2014). TIDE can deconvolve the sequencing chromatograms to detect the fractions of edited and non-edited species. Notably, no unedited Nek10 DNA was detected by TIDE in the Nek10 KO PCR sample. These results are now described in subsections “Nek10 auto-phosphorylates on serine and tyrosine residues in vitro and in cells” and “In vitro kinase assays” of the revised text. Therefore, we can firmly conclude that all Nek10 loci are edited, and the Nek10 KO cells can only produce out-of-frame Nek10 transcripts.

In addition to this gDNA analysis, we also followed the reviewers’ suggestion and quantified Nek10 transcript levels by qRT-PCR. We could not detect any decrease in Nek10 mRNA levels in the Nek10 KO cells compared to the parental U-2 OS. However, not all out-of-frame transcripts are directly degraded by nonsense-mediated decay, and loss of protein expression is therefore not necessarily accompanied by loss of mRNA levels (see for example Dabrowska et al., 2018). To have a closer look at what happens on the mRNA level, we then reverse-transcribed the mRNA from parental and Nek10 KO cells, and performed a PCR spanning the edited region. The results of this experiment are shown in Author response image 1 for the reviewers’ appreciation. A clear single band can be observed for the parental cells, while the Nek10 KO cells contain multiple bands. Hence, the editing we observe on the genomic level, results in the expression of aberrant transcripts. Taken together, we therefore, conclude that the Nek10 KO cells do not contain unedited Nek10 transcripts.

**Author response image 1. respfig1:** RNA was harvested from WT U-2 OS cells (parental) or a Nek10 KO U-2 OS clone (Nek10 KO), and reverse transcribed to generate cDNA. A PCR with primers spanning the Nek10 sgRNA target locus was performed and analyzed by electrophoresis.

In any case, the conclusions drawn on basis of the results presented in Figure 5 will not change even in the unlikely event that functional Nek10 protein is translated from the aberrant transcripts. We generated the KO cell-lines to prevent in trans auto-phosphorylation by endogenous Nek10, but as can be observed in Figure 5E, endogenous Nek10 is not abundant or active enough in these U-2 OS cells to detectably phosphorylate the ectopically expressed kinase-dead Nek10.

Lastly, we apologize for the lack of clarity considering the use of wild-type or Nek10 KO U-2 OS cells. We have clarified this in the Results section and in the legend for Figure 5.

5) The identification of Y644 as the major site of tyrosine phosphorylation of Nek10 (Figure 5E) is compromised by the knowledge that this mutation at this site decreases kinase activity (Figure 5G) – could the decrease in phosphorylation of the Y644 mutant simply result from decreased protein kinase activity rather than Y644 being the major site of phosphorylation? A similar criticism could be levelled at the observed effect of the Y644 mutation in the trans phosphorylation study (Figure 5F) if the effect of the Y644 mutation was to cause decrease phosphorylation associated with a conformational change that alters the activity of Nek10. Also, the conclusion that Nek10 autophosphorylation can occur in trans in vitro is demonstrated (Figure 5F) and it is reasonable to assume that this phosphorylation in trans may occur in cells, but there is no data to support this conclusion (subsection “Nek10 auto-phosphorylates on serine and tyrosine residues in trans, both in vitro and in cells”). The data and the conclusion supporting this part of the study should be discussed with these points taken into account.

The reviewers are correct to point out that the decrease in tyrosine phosphorylation of the Nek10 Y644F variant that we observed in Figure 5E could be caused both by reduced activity of the Y644F mutant, as well as by reduced auto-phosphorylation of the Y644 target site. Also, the reviewers are correct to point out that in trans auto-phosphorylation is demonstrated in vitro, but not necessarily in cells. For both points we have adapted the Results section of Figure 5 (subsection “Nek10 auto-phosphorylates on serine and tyrosine residues in vitro and in cells”) to reflect these more correct interpretations. However, the data depicted in Figure 5F cannot be explained by loss of kinase activity due to the Y644F mutation, as the reviewers suggest, but instead are strong evidence that Y644 is a direct target site in vitro for in trans auto-phosphorylation. To clarify: the FLAG-tagged kinase used in lane 2 and lane 3 is in both cases a tyrosine-specific Nek10 variant, without any additional mutations. The HA-tagged Y644 control (lane 2) or the Y644F mutation (lane 3) was made in the kinase-dead Nek10 mutant that serves as a substrate for in trans auto-phosphorylation by the FLAG-tagged tyrosine-specific Nek10 variant to specifically avoid the potential complication of any Y644F effects on activity. We have adapted the Results section (subsection “Nek10 auto-phosphorylates on serine and tyrosine residues in vitro and in cells”) to explain this experiment more clearly.

6) Figure 7E and F report that both Plk1 and Nek7 can phosphorylate the Cdc27 and RacGAP1 proteins. It is also stated that mutation of residues that were putative phosphorylation sites for both Plk1 and Nek7 led to reduced phosphorylation. This reduction in phosphorylation should be quantified. In addition, it would help the conclusions if the wild-type substrates (or phosphorylated fragments) were subject to mass spectrometry analysis after incubation with the two kinases to demonstrate that there is significant overlap in the phospho-acceptor sites.

We thank the reviewers for this suggestion, and have added the quantification of the Cdc27 and RacGAP1 phosphorylation to Figures 7E and 7F. As can be appreciated, mutation of the target sites on Cdc27 clearly reduced phosphorylation both by Nek7 and Plk1. This is however not the case for RacGAP1, indicating that both kinases target other sites in addition to S170. We had already indicated this conclusion in the original manuscript, but have made it even more explicit in the third paragraph of subsection “Nek6/7/9 share a common substrate motif with Polo-like Kinase 1 and can phosphorylate the same substrates” of the revised manuscript to avoid any misinterpretation.

We appreciate the reviewers’ suggestion to perform a mass spectrometry analysis of Cdc27/RacGAP1 after Plk1/Nek7 phosphorylation. However, we would like to point out that the substrate was generated by IVTT for Figure 7, which does not generate sufficient amounts of substrate for mass spectrometry analysis. Hence, the experiment suggested by the reviewers would be a significant effort involving production of recombinant RacGAP1 and Cdc27, both of which are large proteins and likely difficult to express in bacteria, and would involve multiple mass spectrometry repeats to obtain quantitative information. We do not have the resources or funding required to perform these experiments, and would ask the reviewer for leniency here. In summary, we show in Figure 7 that (1) Nek6/7/9 and Plk1 have extremely similar motifs, (2) there are many phosphosites in both the whole proteome and a Plk1 phosphoproteomic dataset that would be very good substrates for both Nek7 and Plk1, and (3) two Plk1 substrates are also phosphorylated by Nek7, and in the case of Cdc27 even on the same site. We believe this is sufficient evidence to support the conclusions made in the manuscript.

References:

Dabrowska M, Juzwa W, Krzyzosiak WJ, Olejniczak M. Precise Excision of the CAG Tract from the Huntingtin Gene by Cas9 Nickases. Front Neurosci. 2018; 12: 75.

doi: 10.3389/fnins.2018.00075